# Interferon-driven brain phenotype in a mouse model of RNaseT2 deficient leukoencephalopathy

Matthias Kettwig [1,18✉], Katharina Ternka [1,18], Kristin Wendland[1], Dennis Manfred Krüger[2], Silvia Zampar [3], Charlotte Schob[1], Jonas Franz [4,5,6], Abhishek Aich [7], Anne Winkler[4], M. Sadman Sakib[2], Lalit Kaurani[2], Robert Epple[2], Hauke B. Werner [8], Samy Hakroush[9], Julia Kitz[9], Marco Prinz [10,11,12], Eva Bartok [13,14], Gunther Hartmann[13], Simone Schröder[1], Peter Rehling[7], Marco Henneke[1], Susann Boretius [15], A. Alia[16,17], Oliver Wirths [3], Andre Fischer[2,3], Christine Stadelmann [4], Stefan Nessler[4,19] & Jutta Gärtner[1,19]

Infantile-onset RNaseT2 deficient leukoencephalopathy is characterised by cystic brain lesions, multifocal white matter alterations, cerebral atrophy, and severe psychomotor impairment. The phenotype is similar to congenital cytomegalovirus brain infection and overlaps with type I interferonopathies, suggesting a role for innate immunity in its pathophysiology. To date, pathophysiological studies have been hindered by the lack of mouse models recapitulating the neuroinflammatory encephalopathy found in patients. In this study, we generated *Rnaset2$^{-/-}$* mice using CRISPR/Cas9-mediated genome editing. *Rnaset2$^{-/-}$* mice demonstrate upregulation of interferon-stimulated genes and concurrent IFNAR1-dependent neuroinflammation, with infiltration of CD8$^+$ effector memory T cells and inflammatory monocytes into the grey and white matter. Single nuclei RNA sequencing reveals homeostatic dysfunctions in glial cells and neurons and provide important insights into the mechanisms of hippocampal-accentuated brain atrophy and cognitive impairment. The *Rnaset2$^{-/-}$* mice may allow the study of CNS damage associated with RNaseT2 deficiency and may be used for the investigation of potential therapies.

[1] Department of Pediatrics and Adolescent Medicine, Division of Pediatric Neurology, University Medical Center Göttingen, Georg August University, Göttingen, Germany. [2] Department for Epigenetics and Systems Medicine in Neurodegenerative Diseases, German Center for Neurodegenerative Diseases (DZNE), Göttingen, Germany. [3] Department of Psychiatry and Psychotherapy, University Medical Center Göttingen, Georg August University, Göttingen, Germany. [4] Institute of Neuropathology, University Medical Center Göttingen, Georg August University, Göttingen, Germany. [5] Campus Institute for Dynamics of Biological Networks, University of Göttingen, Göttingen, Germany. [6] Max Planck Institute for Experimental Medicine, Göttingen, Germany. [7] Department of Cellular Biochemistry, University Medical Center Göttingen, Georg August University, Göttingen, Germany. [8] Department of Neurogenetics, Max Planck Institute of Experimental Medicine, Göttingen, Germany. [9] Institute of Pathology, University Medical Center Göttingen, Georg August University, Göttingen, Germany. [10] Institute of Neuropathology, Faculty of Medicine, University of Freiburg, Freiburg, Germany. [11] Signalling Research Centres BIOSS and CIBSS, University of Freiburg, Freiburg, Germany. [12] Center for Basics in NeuroModulation (NeuroModulBasics), Faculty of Medicine, University of Freiburg, Freiburg, Germany. [13] Institute of Clinical Chemistry and Clinical Pharmacology, University Hospital, University of Bonn, Bonn, Germany. [14] Unit of Experimental Immunology, Department of Biomedical Sciences, Institute of Tropical Medicine, Antwerp, Belgium. [15] Functional Imaging Laboratory, German Primate Center, Leibniz Institute for Primate Research, Göttingen, Germany. [16] Institute for Medical Physics and Biophysics, University of Leipzig, Leipzig, Germany. [17] Leiden Institute of Chemistry, Leiden University, Leiden, The Netherlands. [18] These authors contributed equally: Matthias Kettwig, Katharina Ternka. [19] These authors jointly supervised this work: Stefan Nessler, Jutta Gärtner. ✉email: matthias.kettwig@med.uni-goettingen.de

Human RNaseT2 deficiency is a rare, monogenetic, early childhood onset cystic leukoencephalopathy (CLE) which manifests as psychomotor delay, spasticity, epilepsy, and normo- or microcephaly during the first year of life. Brain magnetic resonance imaging (MRI) reveals frontal and temporal lobe cystic lesions, multifocal white matter alterations, and cerebral atrophy[1,2]. The phenotypic features of RNaseT2-deficient CLE are indistinguishable from the sequelae of in utero cytomegalovirus (CMV) brain infection. Moreover, there is a significant clinical and neuroradiological overlap of RNaseT2-deficient CLE with Aicardi-Goutières syndrome (AGS), with some affected individuals demonstrating cerebrospinal fluid (CSF) pleocytosis, elevated levels of CSF neopterin as an inflammatory marker, and an overexpression of interferon-stimulated genes (ISGs) in peripheral blood[3].

Congenital viral brain infections are characterized by increased concentrations of type I interferons (IFN-I) in serum and CSF[4,5]. IFN-I is an essential antiviral cytokine, which induces a broad range of antiviral effector genes and inhibits cellular transcription, translation, and proliferation as well as enhancing cross presentation and driving clonal T cell expansion[6,7]. However, despite its vital function, chronic or constitutive IFN-I release can also cause devastating autoinflammation, particularly in the central nervous system (CNS)[4]. While mice with transgenic constitutive IFN-I expression from astrocytes are exceptionally resistant to neurotropic viruses, they demonstrate progressive inflammatory encephalopathy, including cerebral calcification[8,9]. Similarly, IFN-I in the CSF is a hallmark of AGS and considered the principle driver of the severe autoinflammatory encephalopathy observed in these patients.

IFN release is triggered by the activation of innate anti-viral pattern recognition (immune sensing) receptors (PRR) during viral infection, many of which sense viral nucleic acids[10]. However, other triggers such as the mislocalization or modification of self nucleic acids can also activate these pathways[10,11]. In particular, defects in genes involved in nucleic acid immune sensing, nucleic acid editing, and nucleic acid degradation can lead to inappropriate, chronic IFN-I release, known as type I interferonopathy[12]. AGS is the prototype of this disease, with the clinical hallmarks of severe psychomotor retardation, brain white matter abnormalities, and cerebral calcifications. Affected individuals can also present with extraneurological features, such as hepatosplenomegaly, inflammatory bowel disease, systemic lupus erythematosus and abnormalities of hematopoiesis[13–15].

A number of the genes associated with type I interferonopathy and AGS code for nucleases, including *TREX1*, *RNASEH2A*, *RNASEH2B*, *RNASEH2C*, *DNASE2*, and *SKIV2L*. Hypomorphic variants of these genes lead to defects in nuclease activity and thus the accumulation of self nucleic acids and erroneous activation of anti-viral nucleic acid immune sensing receptors[12,16]. In line with these observations, RNaseT2 deficient CLE is caused by loss-of-function mutations of the *RNASET2* gene, a member of the endoribonuclease family that consists of RNase A, RNase T1, and RNase T2. RNaseT2 is exceptionally well conserved across species, including bacteria, yeast and plants and is involved in the endosomal degradation of RNA from mitochondria and effer-ocytosed cells[17–20]. It has been shown that RNaseT2 activity induces type I interferon signaling by degrading longer exogenous RNA molecules into ligands for the pattern recognition receptor TLR8[21,22].

In the last decade, mouse models for inborn type I interferonopathies, including all currently known AGS genes, have been generated to gain insights into disease mechanisms (Supplementary Table 1). Notably, other than the microglia-specific *Usp18*[−/−] mouse, none of these models were reported to display neuroinflammation or other neuropathologies. In *Usp18*[−/−] mice,

the missing negative regulator of Stat1 signaling leads to a tonically active IFN-I signal that causes a destructive white matter microgliopathy[23]. However, some models (*Trex1*, *Adar1*, and *Ifih1* mutant mice) still demonstrated inflammatory and autoimmune phenotypes outside the CNS, such as myocarditis, inflammatory liver disease, lupus-like nephritis, and other systemic autoimmune symptoms in combination with impaired murine hematopoiesis[24–27]. In contrast, *Samhd1*-deficient mice are healthy, despite displaying an interferon signature in distinct tissues and cells, while *Rnaseh2*-deficient mice generally die on embryonic day 9.5 without inflammatory changes[28,29].

In this work, we present the first murine model of inborn RNaseT2 deficiency. Strikingly, in contrast to almost all other mouse models of type I interferonopathy, *Rnaset2*[−/−] mice demonstrate type I IFN upregulation in the CNS and distinct CNS abnormalities that resemble characteristics of the human disease including brain atrophy and cognitive impairment. Outside the CNS, *Rnaset2*[−/−] mice recapitulate prominent features of known AGS models such as lymphocytic infiltration and the presence of an interferon signature in various organs, disturbed hematopoiesis, and autoantibody formation. As the first described whole-body mouse model of type I interferonopathy with a CNS phenotype, the *Rnaset2*[−/−] mouse provides important insights into the mechanisms of IFN-I-dependent neurodegeneration. Moreover, *Rnaset2*[−/−] mice might contribute to develop novel therapeutic approaches for IFN-mediated CNS inflammation.

## Results

**Endonuclease-mediated generation of *Rnaset2*a[tm1(KOMP)Wtsi] *Rnaset2*b[em1Gaer] double knockout mice (*Rnaset2*[−/−]).** Two orthologues of the single human *RNASET2* gene—*Rnaset2a* and *Rnaset2b*, are present in close proximity on chromosome 17 in the mouse genome. Interestingly, they show minor base pair differences at the DNA level, whereas at the protein level, the two *Rnaset2* orthologues are identical. Due to their proximity, generating double-knockout mice by classical interbreeding of *Rnaset2a* and *Rnaset2b* single-knockout mice was unfeasible[30]. However, using CRISPR/Cas9 mediated genome editing, we were able to generate the RNT2AB-double mutant mice by modifying the *Rnaset2b* locus in the background of the *Rnaset2*a[tm1(KOMP)Wtsi] (RNT2A-KO) (Supplementary Fig. 1a). In brief, the *Rnaset2*[−/−] was generated by injecting CRISPR/Cas9 reagents into oocytes of RNT2A-KO mice. Founder lines were screened for an early stop codon at the end of exon 2 in the *Rnaset2b* locus and for the transgenic cassette (EUCOMM) in the *Rnaset2a* gene by PCR and Sanger sequencing (Supplementary Fig. 1a). Line 7 was backcrossed with C57BL/6N wildtype (WT) mice to generate N1 F0 heterozygous mice and then further expanded. Deficiency of RNaseT2a and b on protein level in the homozygous line *Rnaset2*a[tm1(KOMP)Wtsi]; *Rnaset2*b[em1Gaer] (*Rnaset2*[−/−]) was confirmed by immunoblot analysis of embryonic fibroblasts (MEFs) (Supplementary Fig. 1b). WT, heterozygous (*Rnaset2*[+/−]) and homozygous (*Rnaset2*[−/−]) mice were born in the expected Mendelian ratio and homozygous *Rnaset2*[−/−] mice were viable and fertile.

**Rnaset2[−/−] mice have a reduced life expectancy.** At the age of weaning at 3 weeks, *Rnaset2*[−/−] mice were indistinguishable from heterozygous or wild-type siblings regarding their feeding and social behavior, nesting instinct and activity (not quantified). At ~10 weeks of age, *Rnaset2*[−/−] mice displayed an enlarged abdominal girth due to hepatosplenomegaly. At this age, most *Rnaset2*[−/−] mice showed slightly reduced activity but no evidence of pain. For animal welfare reasons the *Rnaset2*[−/−] mice were frequently scored by animal caretakers and veterinarians with

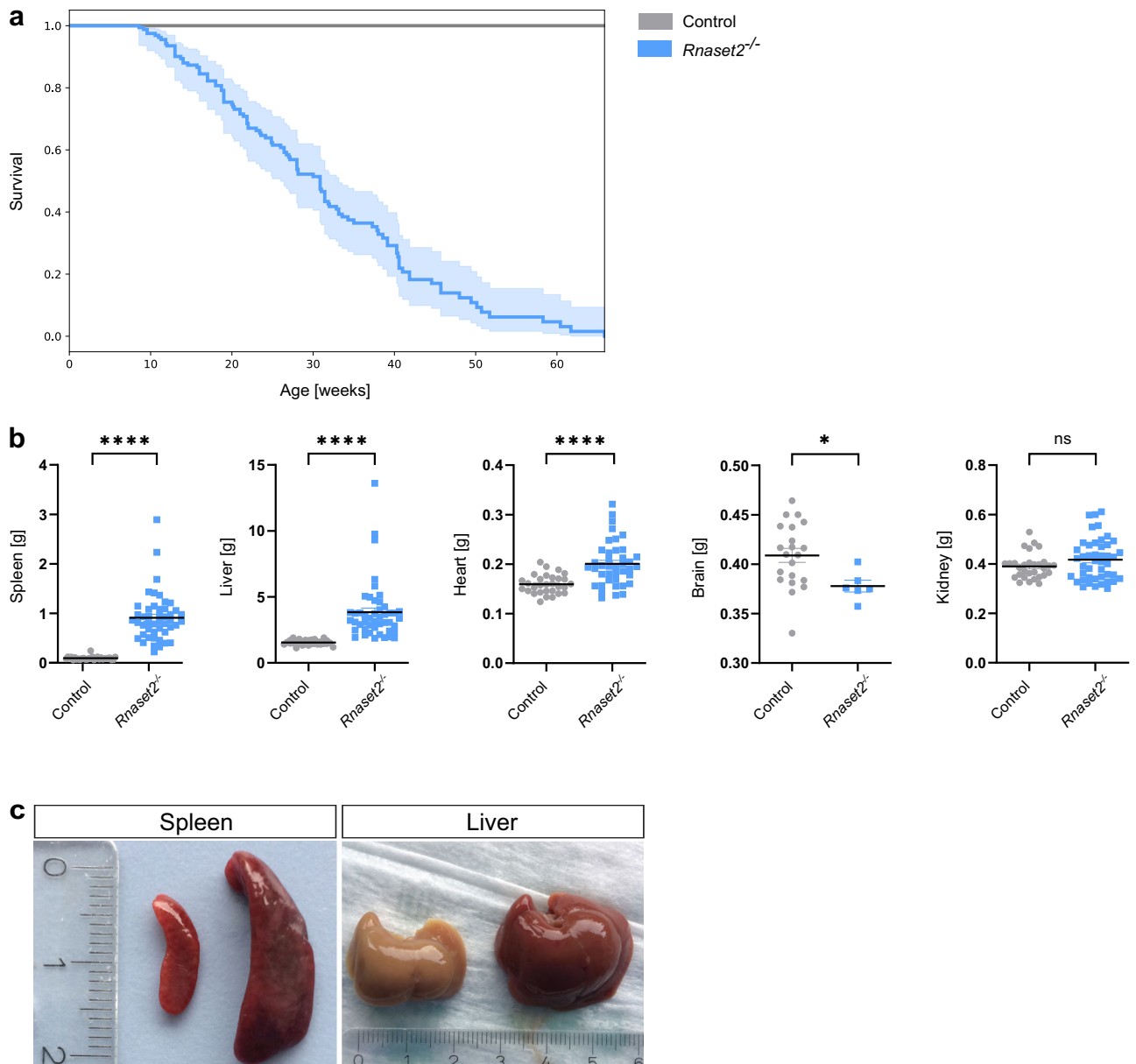

**Fig. 1 Survival and organ enlargement of Rnaset2$^{-/-}$ mice. a** Kaplan–Meier blot survival for Rnaset2$^{-/-}$ (blue, $n = 173$) and control (gray, $n = 102$) mice. The median lifetime of Rnaset2$^{-/-}$ mice is 31 weeks. For the survival curve, a confidence interval of 95% is estimated using the exponential Greenwood method. The alpha level was corrected using Bonferroni's method by the number of comparisons (alpha = 0.05/3). **b** Comparison of organ weights from Rnaset2$^{-/-}$ mice ($n = 47$ for spleen and liver, $n = 40$ for heart, $n = 45$ for kidney and $n = 6$ for brain) and from control mice ($n = 30$ for spleen, liver, heart and kidney and $n = 21$ for brain). Rnaset2$^{-/-}$ mouse revealed a significant enlargement of spleen, liver, and heart ($p < 0.0001$, respectively). Brain weight was significantly reduced ($p = 0.0336$) and kidney remained unaltered ($p = 0.1205$). Data are represented as mean values ± SEM and p values are depicted as ****$p < 0.0001$, *$p < 0.05$ and not significant $p \geq 0.05$. **c** Macroscopy picture of Rnaset2$^{-/-}$ and control spleen (left) and liver (right) after PBS perfusion in 5-month-old representative mice. Rnaset2$^{-/-}$ spleen and liver were enlarged in volume and showed a dark red color. Source data are provided as a Source Data file.

a score sheet adapted to this particular animal model. The reporting of animals reaching the predefined endpoint criteria was genotype-blinded. The determined median lifetime of Rnaset2$^{-/-}$ mice was 31 weeks (Fig. 1a).

Upon autopsy, organ weight analysis of Rnaset2$^{-/-}$ mice revealed approximately 10-fold enlarged spleens ($0.91 \pm 0.48$ vs. $0.09 \pm 0.03$ g), 2.5-fold enlarged livers ($3.84 \pm 2.19$ vs. $1.55 \pm 0.19$ g) and moderately enlarged hearts ($0.20 \pm 0.04$ vs. $0.16 \pm 0.02$ g), whereas their brains were significantly decreased in weight ($0.38 \pm 0.01$ vs. $0.41 \pm 0.03$ g). The kidney weight remained unchanged (Fig. 1b). Notably, liver and spleen had a dark red

color despite extensive perfusion with phosphate-buffered saline (PBS) (Fig. 1c), indicative of extramedullary hematopoiesis.

**Rnaset2$^{-/-}$ mice develop autoinflammatory organ disease and disturbed hematopoiesis.** In line with the dark red color observed (Fig. 1c), the spleen of Rnaset2$^{-/-}$ mice showed massive extramedullary hematopoiesis (Supplementary Fig. 2a–e). Furthermore, histological evaluations demonstrated an increased white splenic pulp (Fig. 2a) and predominantly lymphocytic infiltrates in the heart and liver (Fig. 2b, c) of mutant mice.

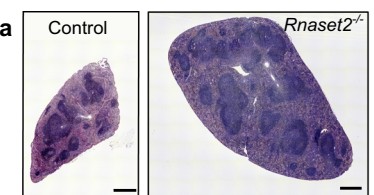

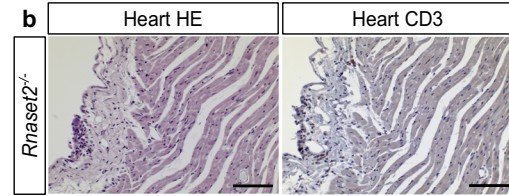

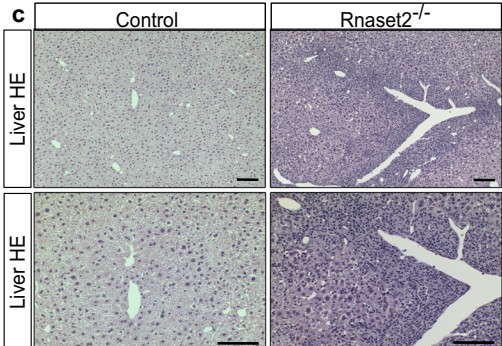

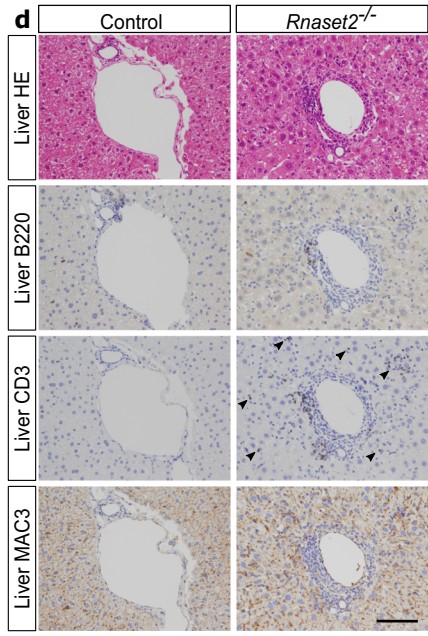

**Fig. 2 Inflammatory organ disease. a** Stitching of haematoxylin-eosin (HE) stained whole spleen cross section illustrates the enlargement and disorganisation of white splenic pulp in a 5-month-old mouse. ×20 magnification. Scale bar, 500 μm. **b** HE staining of the heart showed lymphocytic infiltrates in the pericardium. Immunohistochemical staining revealed most of them as CD3+ T cells. Scale bar, 100 μm. **c** Histological HE staining of slides from PFA-fixed and paraffin-embedded liver. Inflammatory lymphocytes infiltrate around the periportal fields pointing to a hepatitis. ×10 and ×20 magnification. Scale bar, 100 μm. **d** Immunohistochemical staining of periportal infiltrating cells revealed presence of B220+ B cells, CD3+ T cells and activated macrophages (MAC3 staining). Notice, T cells were spread over the whole liver parenchyma (arrowheads). ×20 magnification. Scale bar, 100 μm. Images are representative of seven independent experiments.

Immunohistochemical staining identified CD3+ T cells, B220+ B cells and MAC3+ macrophages as the infiltrating cells in the periportal zone of the liver (Fig. 2d). T cells were found throughout the liver parenchyma (Fig. 2c, d). The inflammatory infiltrates in the pericardium of the heart also consisted of CD3+ T cells (Fig. 2b).

Peripheral blood count and blood smears revealed leukocytosis and macrocytic hyperchromic anemia in mutant mice. Moreover, $Rnaset2^{-/-}$ mice showed reticulocytosis, illustrated by a higher number of blue stained enucleated erythrocytes compared to the control animals. Furthermore, severe thrombocytopenia was observed (Fig. 3a, b). Evaluation of bone marrow smears and immunohistochemical analysis of CD41+ megakaryocytes in spleen and bone marrow revealed increased numbers and morphological alterations (dysplasia) of megakaryocytes, demonstrated by single, small and round nuclei and megakaryocyte accumulation (Fig. 3c–e). In addition, $Rnaset2^{-/-}$ mice displayed significantly increased ALT values (185 ± 109 vs. 33 ± 9 U/l), elevated hemolysis parameters (LDH 1624 ± 676 vs. 349 ± 221 U/l and uric acid concentrations 6.0 ± 1.2 vs. 4.2 ± 1.2 mg/dl), while overall protein levels were decreased (4.0 ± 0.8 vs. 5.0 ± 0.3 g/dl) (Fig. 3f).

In line with its enlarged size, FACS analysis of spleen single cell suspensions (splenocytes) revealed a 5.1-fold increase in total cell count (Supplementary Fig. 2a). Multi-color flow cytometric analysis

of hematopoietic precursors in the spleen demonstrated an elevated number of Ter119+ cells, indicating increased hematopoiesis of the erythroid lineage (Supplementary Fig. 2b). Specifically, we found an increased number of all HSC-dependent multipotent stem cells (common myeloid progenitor, granulocyte–macrophage progenitor and megakaryocyte–erythroid progenitor). In addition, all subsequent precursor cells of the myeloid and erythroid lineage were increased, with pre-megakaryocyte-erythroid (Pre-Meg-E) being the only exception (Supplementary Fig. 2c). In erythroid differentiation, we found a shift to immature erythroblasts (disproportional increase of R2 versus R3-R5) (Supplementary Fig. 2d, e).

Flow cytometric analysis of lymphocytes demonstrated that both absolute CD3+ T cell and CD19+ B cell numbers per spleen were significantly increased (Supplementary Fig. 3a). Likewise, the total numbers of both CD4+ and CD8+ T cells per spleen were significantly increased, and the CD4/CD8 ratio was significantly reduced (Supplementary Fig. 3b). Of note, the proportion of CD8+ T cells harboring an effector memory phenotype was significantly increased while the proportion of naïve CD8+ T cells was significantly decreased (Supplementary Fig. 3c). The percentage and the absolute number of Ly6C$^{hi}$ CCR2+ monocytes were increased (Supplementary Fig. 3d).

Previous reports demonstrated an association of interferonopathies with autoimmunity and systemic lupus erythematosus in particular[12,31,32]. Consistent with other interferonopathies, we

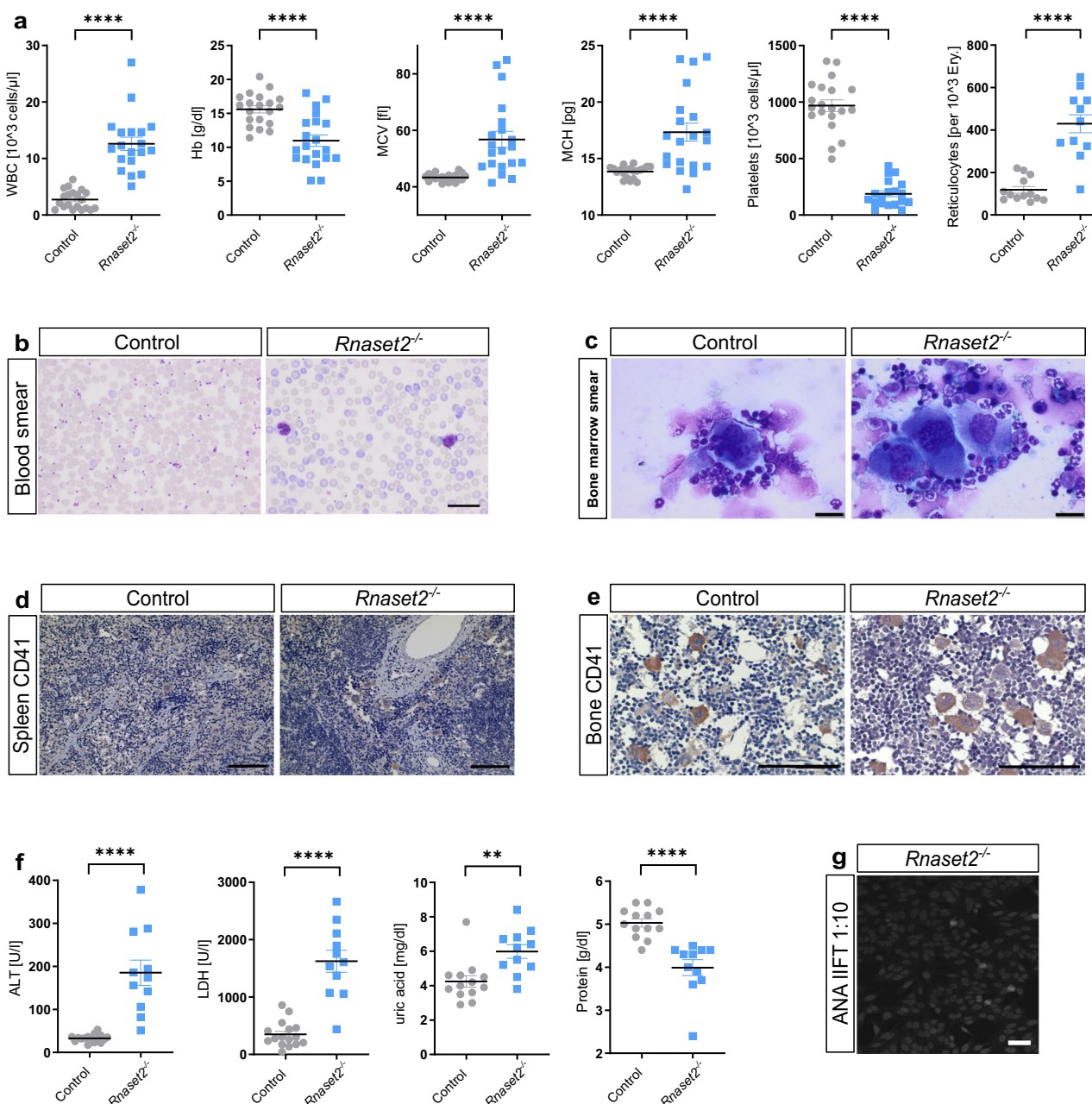

**Fig. 3 Disturbed hematopoiesis and anti-nuclear antibody detection. a** Blood counts from EDTA-stabilized whole blood of *Rnaset2*−/− and control mice revealed increased numbers of white blood cells (WBC, *n* = 19 *Rnaset2*−/− and *n* = 20 control animals) and reticulocytes (*n* = 13 *Rnaset2*−/− and control animals, respectively) in *Rnaset2*−/− mice accompanied by highly significantly reduced platelets (*n* = 19 *Rnaset2*−/−and *n* = 20 control animals) and hemoglobin (Hb) levels (*n* = 20 *Rnaset2*−/− and control animals, respectively). The mean corpuscular volume (MCV) and mean corpuscular hemoglobin (MCH) were significantly elevated (*n* = 20 *Rnaset2*−/− and control animals, respectively). Data depicted as the mean ± SEM. *p* values of two-tailed Student's *t*-test are represented as ****p < 0.0001. **b** Blood smears of *Rnaset2*−/− and control mice clearly demonstrated the severe thrombocytopenia and the massive reticulocytosis (purple cells) in three biological replicates for control and *Rnaset2*−/−, respectively. Scale bar, 25 μm. **c** Bone marrow smear and immunohistochemical staining of **d** spleen and **e** bone marrow showed a higher density of CD41+ megakaryocytes, which presented dysplastic features demonstrated by single, small, and round nuclei and megakaryocyte agglomerations. Spleen and bone marrow images are representative of three independent experiments, scale bar, 100 μm. Bone marrow smear images are representative of four biological replicates for control and *Rnaset2*−/−, respectively, scale bar, 20 μm. **f** Depicted values of clinical chemistry from serum samples of *Rnaset2*−/− (*n* = 11 animals) and control (*n* = 16 animals for ALT and LDH and *n* = 13 for uric acid and protein) mice were significantly altered, whereas further parameters measured remained unremarkable. Data depicted as the mean ± SEM. *p* values of two-tailed Student's *t*-test are represented as **p = 0.0027, ****p < 0.0001. ALT alanine aminotransferase, LDH lactate dehydrogenase. **g** Representative fluorescence microscopy image with a positive antinuclear antibody (ANA) indirect immunofluorescence test (IIFT) of *Rnaset2*−/− mouse serum (12 weeks old) with a titre of 1:10. Further titration even revealed a titre of 1:1000 for autoantibodies present in this serum. Scale bar, 50 μm. *p* values are represented as ****p < 0.0001, ***p < 0.001, **p < 0.01, *p < 0.05 and not significant *p* ≥ 0.05. Source data are provided as a Source data file.

found higher levels of anti-nuclear antibodies in the sera of $Rnaset2^{-/-}$ mice compared to controls (Fig. 3g). Anti-nuclear antibodies were already present at the age of 6 weeks but did not lead to immune complex-mediated nephritis during the observation period of 6 months (Supplementary Fig. 4).

**$Rnaset2^{-/-}$ mice show a strong interferon-stimulated gene response in multiple organs.** We next analyzed ISG transcripts by qRT-PCR in various organs of $Rnaset2^{-/-}$ mice. We observed the highest increase of ISG transcripts in the brain, where the mRNA-abundance of *Isg15*, *Usp18* and *Cxcl10* was elevated 21-fold, 23-fold, and 20-fold, respectively. Moreover, *Ifi27* and *Ddx58* were increased in the brain of $Rnaset2^{-/-}$ mice 3.4-fold and 6.44-fold, respectively (Supplementary Fig. 5a). In the liver, *Ifi27* (8.2-fold), *Rsad2* (6.8-fold), and *Siglec1* (11.3-fold) were elevated (Supplementary Fig. 5b). In the spleen, *Isg15* and *Rsad2* were increased 9.4-fold while *Ifi27* and *Irf7* displayed about 2.5-fold increased mRNA-abundance (Supplementary Fig. 5c). Altogether, $Rnaset2^{-/-}$ mice display a strong interferon response in several organs, most prominently in the brain.

**$Rnaset2^{-/-}$ mice develop neuroinflammation and brain atrophy.** Histology of $Rnaset2^{-/-}$ mouse brain and spinal cord revealed focal immune cell infiltration, including CD3$^+$ T cells, MAC3$^+$ macrophages/activated microglia, and polymorpho-nuclear granulocytes (Fig. 4a). CD3$^+$ lymphocytes were predominantly CD8$^+$ and presented in a perivascular and diffusely infiltrating manner (Fig. 4a, b). The focal inflammatory lesions were accompanied by monocyte infiltration and microglia activation and affected the grey and white matter of all brain regions and the spinal cord (Fig. 4c). In particular, cortex and hippocampus were affected by diffusely infiltrated CD8$^+$ T cells (Fig. 4a). Myelin immunohistochemistry did not reveal any evidence for demyelination (Supplementary Fig. 6).

Flow cytometric ex vivo analysis of the CNS inflammatory infiltrates in $Rnaset2^{-/-}$ mice was consistent with histopathology and demonstrated significantly increased numbers of CD8$^+$ T cells (CD45$^+$ TCRβ$^+$ CD8$^+$) and significant but numerically less impressive increases in CD4$^+$ T cells (CD45$^+$ TCRβ$^+$ CD4$^+$), B cells (CD45$^+$ CD19$^+$) and inflammatory monocytes (CD45$^+$ CD11b$^+$ Ly6C$^{hi}$ CCR2$^+$) compared to controls (Fig. 4d). Of note, infiltrating CD8$^+$ T cells predominantly harbored an effector memory phenotype (CD45$^+$ TCRβ$^+$ CD8$^+$ CD44$^{hi}$ CD62L$^-$) (Fig. 4e and Supplementary Fig. 7).

The immunohistologically proven neuroinflammation in $Rnaset2^{-/-}$ mice was confirmed by contrast-enhanced cerebral magnetic resonance imaging (MRI) revealing significant leakage of the blood-brain barrier for gadopentetate-dimeglumine (Gd-DTPA) (Fig. 5a).

Volumetric MRI of the hippocampi at the age of 4–6 months demonstrated significant hippocampal atrophy in $Rnaset2^{-/-}$ mice compared to controls (Fig. 5b, c). A significant increase in T$_2$ relaxation time in the periventricular region, hippocampus and cortex of $Rnaset2^{-/-}$ mice was observed compared to controls, a well accepted MRI marker for inflammation (Fig. 5d)[33]. Furthermore, ventricles of $Rnaset2^{-/-}$ mice were enlarged at 4–6 months (Fig. 5b, e), indicating brain volume loss.

**$Rnaset2^{-/-}$ mice exhibit learning difficulties.** To assess whether neuroinflammation and neurodegeneration impair cognitive functions in $Rnaset2^{-/-}$ mice the cross maze task was employed to measure alternation frequency as an indicator of spatial working memory. $Rnaset2^{-/-}$ mice showed a trend towards reduced alternation rates ($31.7 \pm 11.8\%$ vs. $39.2 \pm 9.7\%$; $p = 0.061$) at the age of 4 months. The total number of arm entries

($24.8 \pm 10.3$ vs. $36.8 \pm 11.2$; $p < 0.001$) and the average distance ($16.0 \pm 4.9$ vs. $24.1 \pm 5.6$ m; $p < 0.001$) were significantly decreased (Fig. 6a). Accordingly, the average speed in the open field paradigm was significantly reduced ($0.05 \pm 0.02$ vs. $0.03 \pm 0.01$ m/s; $p < 0.01$) (Fig. 6b). Basic motor performance, tested by balance beam and inverted grid hanging tasks, was not impaired (Supplementary Fig. 8a, b). Learning and memory were also assessed using the novel object recognition task (NOR). In general, mice without any cognitive impairment spend more time exploring a novel object compared to a familiar object. On day 1, when two identical objects were presented, $Rnaset2^{-/-}$ and control mice equally explored both objects (Fig. 6c). However, on the testing day (day 2), $Rnaset2^{-/-}$ mice spent significantly less time exploring the novel object compared to control mice ($P < 0.05$; Fig. 6d) and showed a general impairment in object discrimination, as also seen in the significantly lower discrimination ratio ($0.15 \pm 0.26$ vs. $0.39 \pm 0.11$; $p < 0.01$) (Fig. 6e). The average exploration time for the two objects on day 1 and day 2 was the same for $Rnaset2^{-/-}$ and control mice (Fig. 6c, d).

**Single nuclei RNA sequencing confirms type I interferon driven neuroinflammation and provides insights into the mechanism of cognitive impairment.** To identify which cell types and pathways are involved in neuroinflammation we performed single nuclei RNA sequencing analysis of the caudate putamen (Fig. 7) and hippocampus (Supplementary Fig. 9). Clustering analysis revealed obvious differences when comparing $Rnaset2^{-/-}$ to control tissue in the caudate putamen (Fig. 7a). Expression changes were particularly evident for a cluster that we identified as microglia/monocytes but also for astrocytes, oligodendrocytes, oligodendrocyte precursor cells (OPC), and three clusters representing neurons (neuronal cluster 2, 3, and 4; Fig. 7a). When we compared the differentially expressed genes amongst $Rnaset2^{-/-}$ and control mice across all cell types, we observed a strong upregulation of gene ontology (GO) biological processes linked to an immune response and specifically to type I interferon signaling (Fig. 7b, upper panel). The induction of the type I interferon-signaling was particularly obvious in the cluster that we had identified as microglia/monocytes and to a lesser extent in endothelial cells, oligodendrocytes, and OPCs (Fig. 7b, lower panel).

Therefore, we decided to analyze the microglia/monocyte cluster in more detail and observed that it consisted of microglia that were specific to either $Rnaset2^{-/-}$ or control mice (Fig. 7c). While popular microglia marker genes such as P2ry12, Tmem119, and Siglec-h were slightly decreased upon activation, expression of C4b, an established marker for reactive microglia was highly upregulated in $Rnaset2^{-/-}$ mice (Fig. 7d)[34]. Two additional clusters were specific to $Rnaset2^{-/-}$ tissue and represented infiltrating monocytes and CD8$^+$ T cells, as indicated by the expression of corresponding marker genes (Tbrc2, Cd8a, Gzmb and Stat4 for CD8$^+$ T cells; Ccr2, Ly6c2, and CD74 for infiltrating monocytes) (Fig. 7d). In addition, a number of interferon-stimulated genes (Isg15, Usp18, Cxcl10, and Ddx58) previously determined in various organs by qPCR were also increased in microglia, infiltrating monocytes and CD8$^+$ T cells of $Rnaset2^{-/-}$ mice (Fig. 7e).

To validate the single nuclei RNA sequencing results on the protein level, we carried out immunostainings of interferon-stimulated gene 15 (ISG15) in the cortex of $Rnaset2^{-/-}$ and control mice. We found an impressive upregulation of ISG15 in various cell types in the cortex of $Rnaset2^{-/-}$ mice (Fig. 7f, left panel). Co-immunostaining revealed an upregulation of ISG15 particularly in astrocytes (GFAP), microglia (IBA1), and endothelial cells (CD31) (Fig. 7f, right panel).

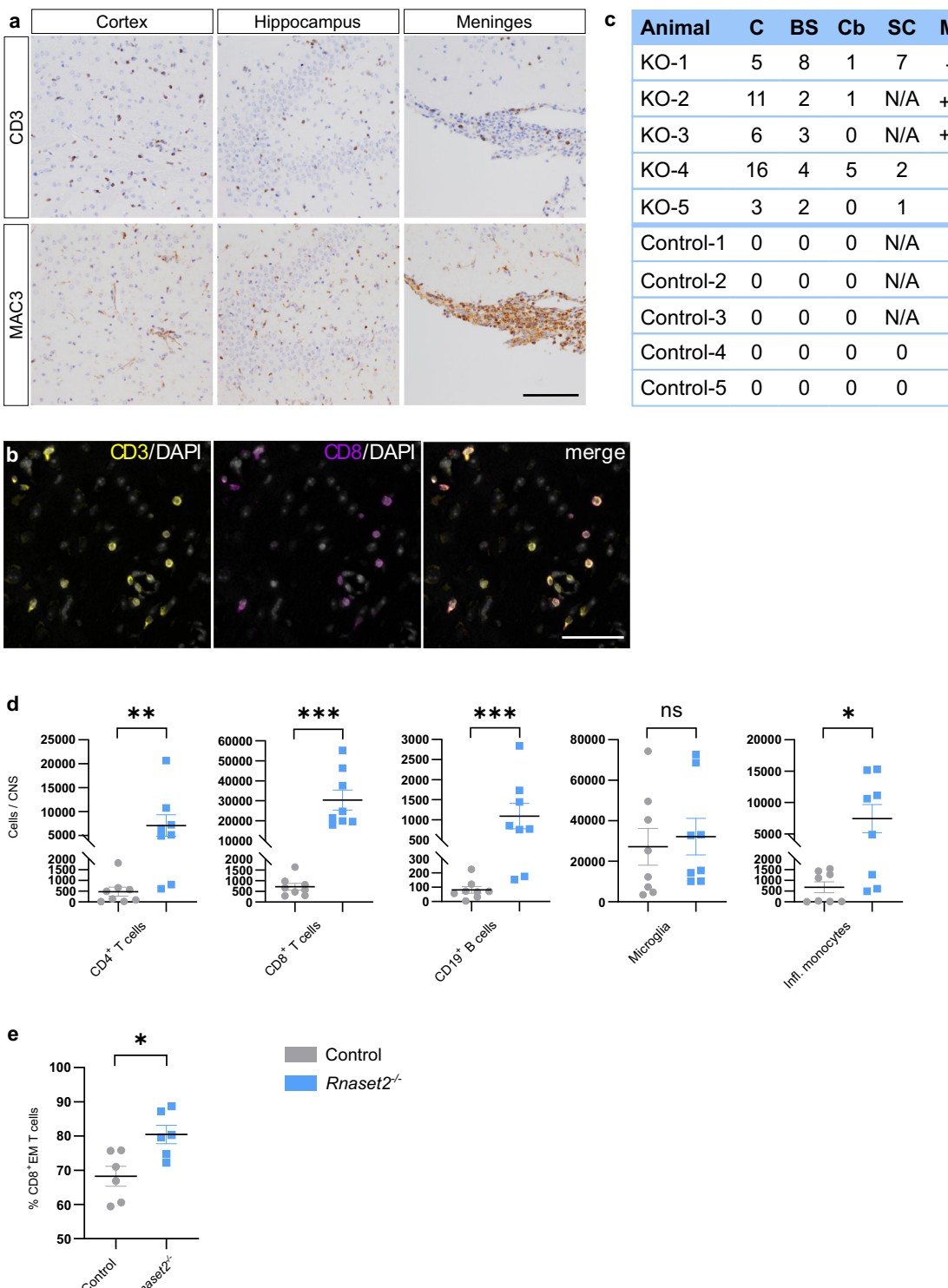

We also analyzed the expression profiles of astrocytes, oligodendrocytes, and oligodendrocyte precursor cells in $Rnaset2^{-/-}$ and control mice (Supplementary Fig. 10). Astrocytes from $Rnaset2^{-/-}$ mice were characterized by a diminished processes associated with neuronal support functions such as ion transport (Supplementary Fig. 10b). GO-analysis furthermore revealed that oligodendrocytes from $Rnaset2^{-/-}$ mice contribute to interferon-mediated inflammation while processes linked to neuronal support such as axon development are reduced (Supplementary Fig. 10c). An induction of inflammation-related processes was also observed in oligodendrocyte precursor cells (Supplementary Fig. 10d).

In addition, we analyzed the neuronal clusters affected in $Rnaset2^{-/-}$ mice. Neuronal cluster 2 showed an obvious difference between $Rnaset2^{-/-}$ and control mice (Supplementary Fig. 10e). The specific expression of the marker genes $Gad1$ and $Gad2$ indicated that these cells mainly represented GABAergic inhibitory neurons (Supplementary Fig. 10f). Differential expression analysis revealed that processes related to energy metabolism and mitochondria function were decreased in this neuronal cluster, which is in line with data suggesting that loss of mitochondrial oxidative phosphorylation precedes neuronal degeneration (Supplementary Fig. 10g)[35–37]. Genes upregulated

**Fig. 4 Neuroinflammation. a** Immunohistochemistry of T lymphocytes (CD3 positive, upper row) and macrophages/ activated microglia (MAC3 positive, lower row) in cortex, hippocampus and meninges of $Rnaset2^{-/-}$ mice revealed diffuse infiltration. Scale bar, 100 μm. **b** Co-immunostaining of CD3$^+$ and CD8$^+$ cells in white matter showed that most infiltrated T cells were CD8 positive. Scale bar, 50 μm. **c** Systematic evaluation of perivascular CD3$^+$ cell infiltrations in standard slices from five $Rnaset2^{-/-}$ and control mice underline the involvement of the whole CNS. Of note, grey and white matter were involved in all regions. C = cerebrum, BS = brain stem, Cb = cerebellum, SC = spinal cord, Mi = meningeal inflammation, N/A = not analyzed. Images in **a**, **b** are representative of these five animals. **d** FACS analysis of total CNS (brain and spinal cord). Data depicted as the mean ± SEM. $p$ values of two-tailed Mann–Whitney test and two-tailed Student's $t$-test (only for microglia) are represented as $*p < 0.05$, $**p < 0.01$, $***p < 0.001$ and not significant $p \geq 0.05$ of $Rnaset2^{-/-}$ (n = 8) and control mice (n = 8). $Rnaset2^{-/-}$ mice revealed significantly increased numbers of CD4$^+$ T-cells (CD45$^+$ TCRβ$^+$ CD4$^+$, $p = 0.0019$), CD8$^+$ T-cells (CD45$^+$ TCRβ$^+$ CD8$^+$, $p = 0.0002$), B cells (CD45$^+$ CD19$^+$, $p = 0.0006$) and inflammatory monocytes (CD45$^+$ CD11b$^+$ Ly6C$^{hi}$ CCR2$^+$, $p = 0.0281$) whereas microglia cells (CD45$^{int}$ CD11b$^{int}$ Ly6C$^-$ Ly6G$^-$) were not significantly different ($p = 0.7014$; data pooled from three independent experiments. **e** Significant elevation ($p = 0.0118$) of CD8$^+$ effector memory T cells (CD45$^+$ TCRβ$^+$ CD8$^+$ CD44$^{hi}$ CD62L$^-$) in $Rnaset2^{-/-}$ mice by FACS analysis, n = 6 $Rnaset2^{-/-}$ and control mice; data pooled from three independent experiments. Data presented as mean ± SEM. $p$ values of two-tailed Student's $t$-test is depicted $*p < 0.05$. The corresponding gating strategy is depicted in Supplementary Fig. 7. Source data are provided as a Source data file.

in this neuronal cluster were mainly linked to a defense response to viruses and type I interferon signaling (Supplementary Fig. 10g). Nuclear transcriptional profiles of neurons related to cluster 3 and 4 in control mice were almost completely absent in $Rnaset2^{-/-}$ mice, indicating a prominent change in their transcriptional profiles. The specific expression of the marker genes $Vglut1$ and $Vglut2$ in cluster 3 suggest that these cells are excitatory glutamatergic neurons, while neuronal cluster 4 consists of parvalbumin positive GABAergic inhibitory neurons (Supplementary Fig. 10f).

A similar analysis of cluster-specific gene ontology (GO) biological processes was carried out for hippocampal tissue (Supplementary Fig. 9). We observed a strong inflammatory response driven primarily by microglia cells (Supplementary Fig. 9a,b). In the hippocampus we saw the same activation of microglia compared to caudate putamen, which is reflected by a decreased expression of popular microglia genes ($P2ry12$ and $Siglec-h$) and the expression of the reactive microglia marker C4b. The upregulation of interferon-stimulated genes ($Isg15$, $Usp18$, $Cxcl10$, and $Ddx58$) was also present in hippocampal microglia cells of $Rnaset2^{-/-}$ mice (Supplementary Fig. 9c). Although few genes were differentially expressed in oligodendrocytes and OPCs, there was a strong induction of the $Dddx60$ gene, which codes for an RNA helicase that promotes RIG-I, MDA5, and LGP2-mediated induction of interferon-signaling (Supplementary Fig. 9d)[38]. These inflammatory responses were associated with the loss of a neuronal cluster representing GABAergic inhibitory neurons and a severe dysregulation of the synaptic plasticity genes in excitatory glutamatergic neurons (Supplementary Fig. 9e,f).

Inflammation and in particular type I interferons are known to increase major histocompatibility complex (MHC) class I expression on many cells[39,40]. Not surprisingly, the MHC class I genes $H2-D1$, $H2-K1$, and $B2M$ were among the top regulated genes in our nuclear RNA sequencing data set and their expression was significantly increased in oligodendrocytes, OPCs, astrocytes, neurons, and microglia (Supplementary Fig. 11a–f). To verify these transcriptional findings, we analyzed the MHC class I expression on microglia cells by flow cytometry, which was significantly increased in $Rnaset2^{-/-}$ compared to control mice (Supplementary Fig. 11g).

In summary, single nuclei RNA sequencing was able to identify type I interferon signaling in all CNS cell types. Furthermore, we found evidence for the dysregulation of homeostatic functions in astrocytes, oligodendrocytes and OPCs along with marked alterations in neuronal subclusters.

**Neuroinflammation and hippocampal atrophy in $Rnaset2^{-/-}$ mice are IFNAR1-mediated.** To investigate whether the CNS

inflammation observed in $Rnaset2^{-/-}$ mice is indeed type I interferon-mediated, $Rnaset2^{-/-}$ mice were crossbred to $Ifnar1^{-/-}$ mice. $Rnaset2^{-/-}$ $Ifnar1^{-/-}$ and age-matched $Rnaset2^{-/-}$ mice, as well as control mice, were sacrificed at month six, and brain tissue was collected for immunohistochemical analysis (Fig. 8a). Whereas the $Rnaset2^{-/-}$ mice presented with the typical foci of mononuclear infiltration consisting of T cells and monocytes in grey and white matter regions, the $Rnaset2^{-/-}$ $Ifnar1^{-/-}$ and control mice did not show any infiltration of peripheral immune cells into the brain (Fig. 8a,b). Furthermore, microglia activation was not observed in $Rnaset2^{-/-}$ $Ifnar1^{-/-}$ and control mice (Fig. 8a). Importantly, inflammatory foci in $Rnaset2^{-/-}$ mice were accompanied by upregulation of ISG15, which was entirely absent in $Rnaset2^{-/-}$ $Ifnar1^{-/-}$ and control mice as demonstrated by the color deconvolution of the DAB-ISG15 signal in slide images of the whole hemispheres (Fig. 8a and Supplementary Fig. 13). These data indicate that the concurrent lack of functional type I interferon signaling completely abrogates the neuroinflammatory phenotype elicited in $Rnaset2^{-/-}$ mice.

In line, the $T_2$ relaxation time in the ex vivo MRI normalized in all analyzed brain areas including the hippocampus (Fig. 8c) and the ventricular enlargement disappeared. Of note, the volumetric analysis of the relative hippocampal area showed no evidence for hippocampal atrophy in $Rnaset2^{-/-}$ $Ifnar1^{-/-}$ mice (Fig. 8d, e).

Abolished type I interferon signaling in $Rnaset2^{-/-}$ $Ifnar1^{-/-}$ animals also reduced the observed organ enlargements of $Rnaset2^{-/-}$ mice and normalized many blood values (Supplementary Fig. 12a–c). In this regard, thrombocytopenia and leukocytosis improved but remained significantly different in $Rnaset2^{-/-}$ $Ifnar1^{-/-}$ animals compared to controls whereas the macrocytic hyperchromic anemia completely resolved (Supplementary Fig. 12c).

Importantly, the lack of IFNAR1 signaling in $Rnaset2^{-/-}$ Ifnar1$^{-/-}$ mice also rescued the limited life expectancy of $Rnaset2^{-/-}$mice (Supplementary Fig. 12d). Taken together, we provide compelling evidence that interferon-signaling is essential for neuroinflammation and neurodegeneration in $Rnaset2^{-/-}$ mice and has a significant impact on the reduced life expectancy in this animal model.

## Discussion
The pathophysiological understanding of congenital viral brain infections and inborn type I interferon-driven diseases has been impaired by the failure of many animal models to reproduce the neurological phenotype, which is a fundamental characteristic of these disorders. In this study, we characterize the newly generated $Rnaset2^{-/-}$ mice, in which the CNS, along with other organs, is prominently affected, and reveal that neuroinflammation and neurodegeneration strongly depend on IFNAR1-signalling.

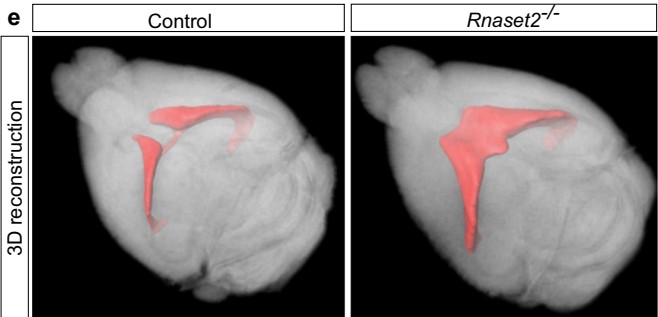

| **a** | IHC: CD3 | MRI: T2w | MRI: Gd-DTPA |

**b** | sagittal | coronal | axial

Control

Rnaset2⁻/⁻

**c** Hippocampus vs. Whole brain area [%]

**d**

| Brain area | T₂ relaxation (ms) | |
|---|---|---|
| | Control | *Rnaset2⁻/⁻* |
| 1 Ventricle | 23.25±0.081 | 23.27±0.118 |
| 2 Periventricular | 21.64±0.241 | 22.28±0.255* |
| 3 Cortex | 21.12±0.415 | 21.91±0.296* |
| 4 Hippocampus | 22.06±0.452 | 23.02±0.683* |

**e** 3D reconstruction | Control | *Rnaset2⁻/⁻*

*Rnaset2⁻/⁻* mice develop a severe type I interferon-driven neuroinflammatory disorder, which is characterized by elevated ISGs, perivascular infiltrates of inflammatory monocytes, activated microglia, and diffuse infiltration of CD8⁺ T cells of the effector memory type throughout the entire CNS. The perivascular inflammation leads to blood-brain barrier leakage as shown by gadolinium enhancement on cerebral MRI. Furthermore, the observed neuroinflammation might foster the documented general brain atrophy, which is accentuated in the hippocampus. Not surprisingly, memory function is impaired in *these* mice.

The first indications of harmful effects of type I interferon on the brain were found in transgenic mice, in which astrocytes

**Fig. 5 Magnetic resonance imaging (MRI) of neuroinflammation and neurodegeneration. a** Light microscopic image of CD3 immunohistochemistry (IHC: CD3) revealed a perivascular T cell infiltration near the left ventricle (inlay, left image). Scale bar, 1 mm and 100 μm respectively. Corresponding T2-weighted coronary MRI (MRI: T2w) showed no remarkable alterations (middle image). Corresponding map of percentage signal enhancement (MRI: Gd-DTPA) revealed gadolinium enhancement in the region of perivascular T cell infiltration near the left ventricle in $Rnaset2^{-/-}$ mice (red circles, right image). Representative images of $n = 2$ biological replicates for $Rnaset2^{-/-}$ and control mice, respectively. **b** Representative magnetic resonance anatomical images (of $n = 8$ $Rnaset2^{-/-}$ and control mice, respectively) in sagittal, coronal, and axial plane showing significant hippocampal atrophy (asterisk) in $Rnaset2^{-/-}$ mice compared to controls. Ventricular enlargement, which may have been arisen due to shrinkage of hippocampus are shown with (arrows). **c** Percent area of hippocampus (left and right) with respect to whole brain area. The size of the hippocampus relative to the whole brain was significantly smaller ($p = 0.0002$). $n = 8$ $Rnaset2^{-/-}$ and control mice from two independent experiments. Data depicted as the mean ± SEM. $P$ value of one-way ANOVA are presented as ***$p < 0.001$. **d** $T_2$ relaxation time measurement in various brain regions displayed a significant increase in cortex, hippocampus and periventricular region of $Rnaset2^{-/-}$ mice in $n = 8$ $Rnaset2^{-/-}$ and control mice, respectively. Region of interest (ROI) selected for $T_2$ relaxation time measurements are shown in left images. ROI: (1) ventricle, (2) periventricular, (3) cortex, (4) hippocampus. Data depicted as the mean ± SEM. $p$ values of one-way ANOVA are represented as *$p < 0.05$. **e** 3-dimensional reconstruction of the ventricular system in MRI showed an expansion in $Rnaset2^{-/-}$ (right side) versus control mice (left side) as an indication of a general brain atrophy. Source data are provided as a Source data file.

chronically secrete IFN-I[8,9]. The reported phenotype, including inflammatory angiopathy with mononuclear cell cuffing, brain atrophy, particularly of the hippocampus, and memory dysfunction resembles the phenotype of the $Rnaset2^{-/-}$ mouse described here. RNaseT2-deficient rats developed a more limited phenotype characterized by reactive astrocytes in the hippocampus and learning difficulties. These rats also showed indications of altered lysosomal function, but lacked cellular evidence of neuroinflammation and type I interferon signatures[41].

The expansion and persistence of effector memory CD8+ T cells is associated with both viral infection and AGS[42–44]. Moreover, in the CNS, adaptive immune cells and clonally expanded effector memory CD8+ T cells have come into focus in the pathogenesis of aging and Alzheimer's disease[45,46]. The observed neuroinflammatory response in $Rnaset2^{-/-}$ mice with considerable numbers of CD8+ T cells exceeds the bystander effects observed in many neurodegenerative disease models[47,48]. It is tempting to speculate that the here reported increase in MHC class I expression on all CNS cells might support antigen-specific CD8+ T cell expansion in the CNS. Thus, clarifying if CD8+ T cells are clonally expanded in the CNS of $Rnaset2^{-/-}$ mice and whether they participate in perpetuating neuroinflammation and neurodegeneration will be an important issue to be addressed in future studies. Understanding the extent to which CD8+ T cells contribute to disease pathogenesis is of great importance and with broad implications for neuroinflammatory and neurodegenerative diseases in general.

As shown by single nuclei RNA sequencing, type I interferon signaling in the CNS, is accompanied by a homeostatic dysfunction of glial cells (microglia, astrocytes, oligodendrocytes, and OPCs) and neurons. In order to improve the mechanistic understanding of type I interferon-driven neuropathological changes, we need to better understand how the observed transcriptional disturbances in glial cells and in neuronal subclusters contribute to the disease pathogenesis.

The unique inflammatory CNS phenotype in $Rnaset2^{-/-}$ mice requires 4–5 months to develop. The lack of CNS inflammation in many AGS mutants described so far might therefore be attributed to their premature death, either embryonically as in $Dnase2a$ mutants or within days postnatally as in $Adar1$ and $Rnaseh2b/c$ mutants[49]. Alternatively, AGS-related genes might have more redundant or dispensable functions in mice than in humans. $Rnaset2^{-/-}$ mice share the described peripheral phenotype with many other AGS mouse models. Interferon signatures in various organs and a reduced survival have been described for $Trex1$, $Adar1$, and $Ifih1$ mutants[25,27,50]. Furthermore, T cells have been shown to infiltrate the heart of $Trex1$-deficient mice, hepatocyte-specific $Adar1$-deficient mice and in gain-of-function mutation $Ifih1$ mutant mice respectively[25,27,51].

In the hepatocyte-specific $Adar1$-deficient mice the infiltrating T cells are also predominantly of the effector memory subtype, suggesting that the clonal expansion of T cells could take place in the effector organ[51]. $Rnaset2^{-/-}$ mice share the disturbance of hematopoiesis, a prominent feature of AGS mouse models, with $Trex1$, $Adar1$, and $Dnase2a$ mutants[26,49,52]. The anemia of $Rnaset2^{-/-}$ mice might recapitulate the hematopoietic disorder of erythroid-restricted deletion of $Adar1$[52]. We found increased ANA autoantibodies in $Rnaset2^{-/-}$ mice, but no evidence for kidney pathology. This is different from $Ifih1$ mutants, which develop an antibody-mediated lupus-like nephritis, or from $Trex1$-deficient mice, which develop lupus-like inflammation in multiple organs[25,44].

IFNAR1-deficiency in $Rnaset2^{-/-}$ mice strikingly abrogates the CNS inflammation observed in $Rnaset2^{-/-}$ IFNAR1-competent mice and prolongs their life expectancy. However, not all perturbations are completely abolished, and hepatosplenomegaly, thrombocytopenia, and leukocytosis are only ameliorated. Thus, future therapeutic approaches against interferonopathies should not only take into account the underlying mechanisms of how nucleic acids activate the interferon-signaling pathway but also the processes that lead to tissue-specific effects of IFN-I. In this regard, we and others unraveled the interplay of TLR8 and RNaseT2 recently. Human RNaseT2 and RNase2 act synergistically to degrade longer, structured RNA molecules into TLR8 ligands. Thus, cells from RNaseT2 hypomorphic patients demonstrated reduced activation of TLR8[21,22]. However, one can assume that type I interferon signaling is not the result of this reduced TLR8 activity, but rather the accumulation of longer RNA molecules that activate other RNA-binding PRRs and trigger interferon production[53]. This would be in line with what has been observed for another endolysosomal nuclease, DNase2, which both contributes to TLR9 activation and inhibits activation of the cytosolic DNA receptors cGAS and AIM2[53–56]. In addition to signaling PRRs, viral restriction factors, such as PKR and OAS1, are ISGs and also sense cytosolic RNA, leading to substantial cell-intrinsic effects. Given that the immune sensing mechanisms activating the interferon pathway are also activated by viral nucleic acids, it is not surprising that the human CNS phenotype of RNaseT2-deficient CLE recapitulates congenital viral infection[1]. However, further studies will be necessary to assess which of the candidate anti-viral RNA receptors in fact contribute to this phenotype.

Altogether, the $Rnaset2^{-/-}$ mouse is an important addition to the repertoire of mouse models for type I interferonopathies. With its unique ability to recapitulate the human neuroinflammatory phenotype of RNaseT2-deficiency, it will provide important insights into the neurological pathomechanisms of human RNaseT2 deficiency. A better understanding of the

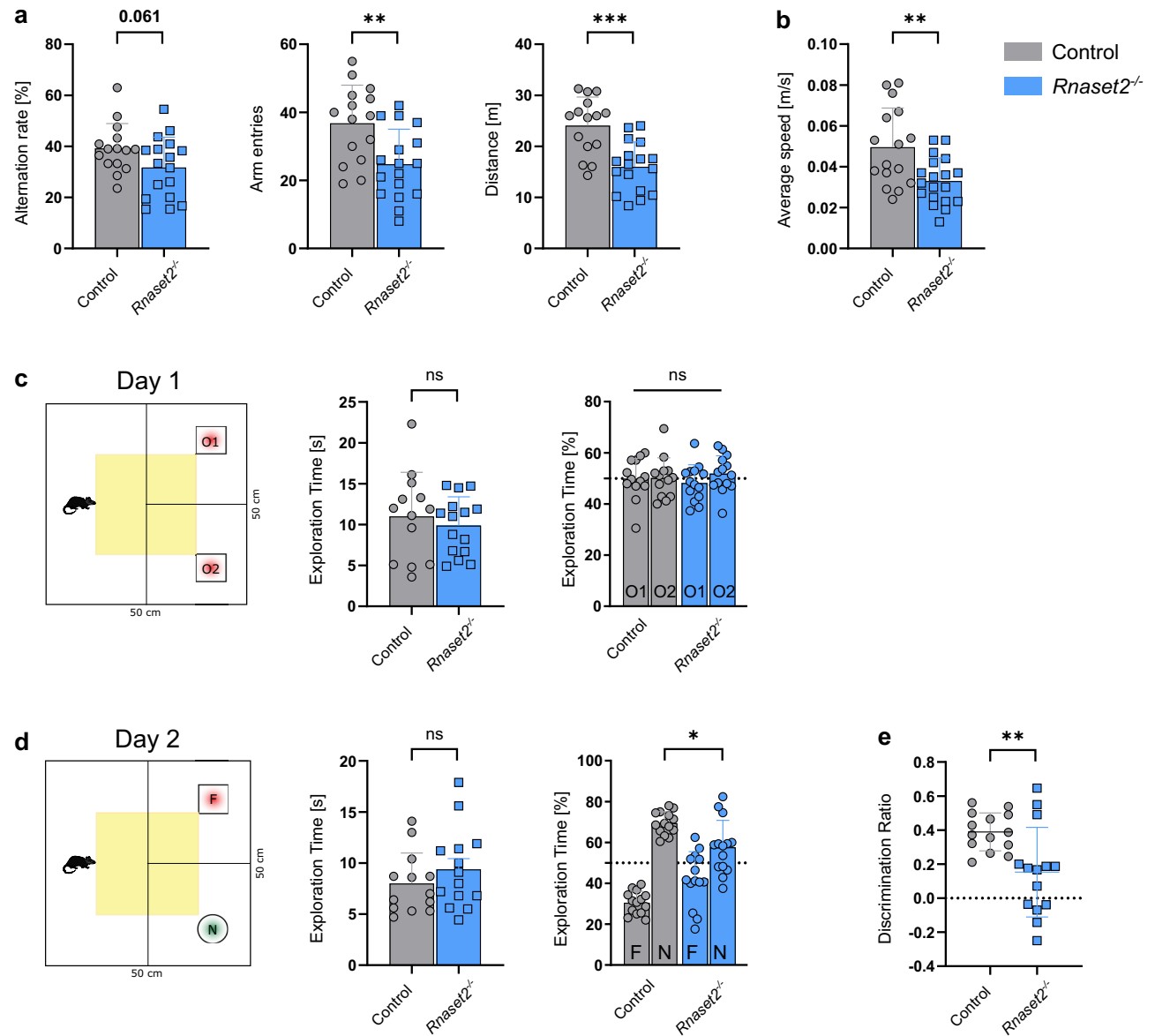

**Fig. 6 Memory impairment in 4-month-old *Rnaset2*$^{-/-}$ mice. a** Cross maze testing: Percentage of spontaneous alternation ($p = 0.061$), total arm entries, and the average distance in meters of *Rnaset2*$^{-/-}$ (blue) and control mice (gray). *Rnaset2*$^{-/-}$ mice showed less activity, documented by a reduced number of arm entries ($p = 0.0034$) combined with a significant decrease of the average distance ($p = 0.0001$) covered in the cross maze. **b** Graph depicting the significantly reduced average speed of *Rnaset2*$^{-/-}$ compared to control mice in the open field box ($p = 0.0031$). **c** Novel object recognition (NOR) analysis in an open field box. Mean percentage of time spent exploring two similar objects (O1 and O2, red) and total exploration time on training day (Day 1) were similar in both genotypes. **d** Mean percentage of time spent exploring two different objects, the familiar (F, red) and a novel one (N, green), on testing day (Day 2) of the NOR test was significantly different between *Rnaset2*$^{-/-}$ mice and controls ($p = 0.0456$). **e** Discrimination ratio of *Rnaset2*$^{-/-}$ mice was significantly diminished ($p = 0.0046$). *Rnaset2*$^{-/-}$ $n = 17$, control $n = 15$. Values are given as the mean ± SEM. Statistical analysis was performed with unpaired t-test and with two-way ANOVA followed by Tukey´s multiple comparison test. $p$ values are represented as ****$p < 0.0001$, ***$p < 0.001$, **$p < 0.01$, *$p < 0.05$ and not significant (ns) $p \geq 0.05$. Source data are provided as a Source data file.

functional consequences of type I interferon-triggered pathways within the CNS will have far-reaching implications not only for inborn RNaseT2-deficient CLE and AGS but also for congenital viral infections. As such, it may be useful to facilitate the design of therapeutic approaches for these severe early childhood onset diseases.

## Methods

**Generation of RNaseT2 knockout mice.** Embryonic stem cells (ES) from a JM8A3.N1 background harboring an engineered allele of the *Rnaset2a* gene were acquired from the European Conditional Mouse Mutagenesis Program (EUCOMM). Chromosomal integration of the transgenic cassette deletes exon 2, 3,

and 4 of *Rnaset2a* [NM_001083938.3] leading to a frameshift and a translational stop in exon 5. ES were microinjected into blastocysts derived from C57BL/6N mice, and embryos were transferred to pseudo-pregnant NMRI foster mothers, yielding 10 chimeric males. For ES clone EPD0405_5_D12, germline transmission was achieved upon breeding with C57BL/6N females, yielding C57BL/6N *Rnaset2a*$^{\text{tm1(KOMP)Wtsi}}$ (RNT2A-KO) mice, which were maintained on a C57BL/6 background. To create the *Rnaset2a* and *b* double knockout strain C57BL/6 N *Rnaset2a*$^{\text{tm1(KOMP)Wtsi}}$; *Rnaset2b*$^{\text{em1Gaer}}$ (*Rnaset2*$^{-/-}$), the knockout of *Rnaset2b* was generated on the background of RNT2A-KO strain by the CRISPR/Cas9 method[57]. We injected the Cas9 protein instead of Cas9 coding plasmid DNA to reduce off-target effects[58]. The CRISPR/Cas9 target sequence was located in exon 2 [5'- ACAGUAUGCAAGGUGAG -3'; crRNA] of the *Rnaset2b* gene [NM_026611.2] and identified with CRISPR guide design tool from the Massachusetts Institute of Technology (http://crispr.mit.edu:8079).

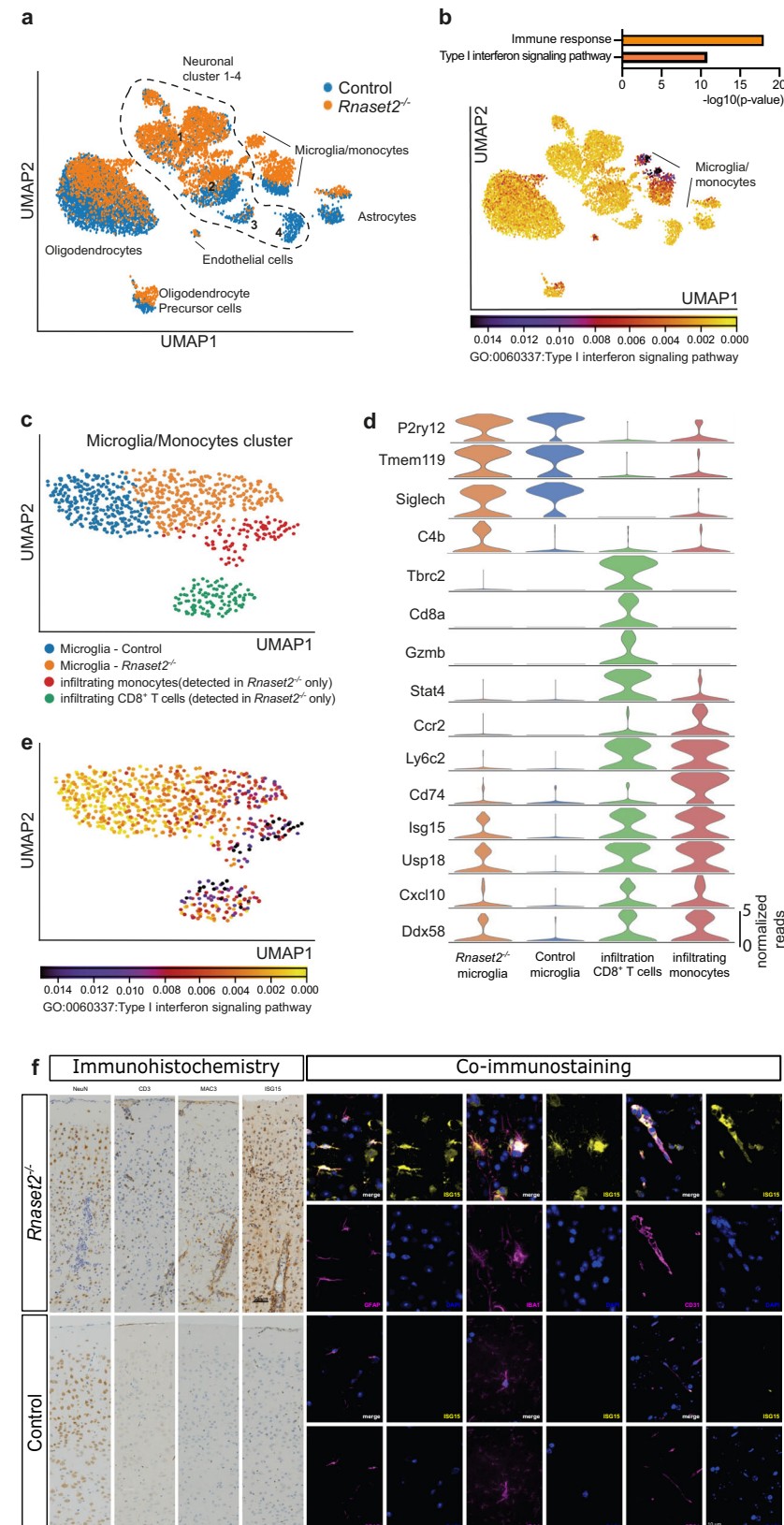

For Ribonucleoprotein- (RNP) mediated genome editing, zygotes of homozygous RNT2A-KO were microinjected with crRNA:tracrRNA:Cas9 RNP-complex (Integrated DNA Technologies Inc., IDT)). In brief, 1.5 µM crRNA was mixed with equimolar tracrRNA to allow the formation of the guideRNA. This was then added to 0.4 µM *S.p.* Cas9 protein (IDT) to form the RNP-complex. A single-stranded template DNA (IDT) was added at last to this injection mix for homology directed repair (HDR) [5'- TTTTGTCACTTTAAAACAGTATTGAAAAATCAAT

TTTTCTTTTAACAGTGGCAGCCATGAGTGGAAAAAACTAATTTTGACCCA GCACTGGTAAGGATCCACAGTATGCAAGGTGAGCTCTGCTTTGACTGGC CCA -3']. The resulting edit in the genome had an insertion of the translational stop codon TAA and a *BamHI* restriction site in exon 2 of *Rnaset2b* (c.145_147delinsTAA; p.Pro49Ter).

Routine genotyping of the *Rnaset2a* knockout allele was performed by genomic polymerase chain reaction (PCR) using primers P1 (5′-TACTTGCAGAAGGCCA

**Fig. 7 Single nuclei RNA sequencing analyses of the caudate putamen confirms type I interferon-driven neuroinflammation. a** UMAP clustering of the caudate putamen from *Rnaset2*$^{-/-}$ and control mice. **b** Upper panel: Differential gene-expression was performed to compare *Rnaset2*$^{-/-}$ and control mice across all detected cell types. GO-term analysis (biological processes) reveals immune response and type I interferon signaling as the major biological processes to be affected in the caudate putamen from *Rnaset2*$^{-/-}$ mice. Lower panel: UMAP indicating that the expression of the type I interferon signaling pathway (GO:0060337) is particularly obvious in the microglia/monocytes cluster. **c** UMAP clustering of the microglia/monocytes cluster clearly distinguished WT and *Rnaset2*$^{-/-}$ microglia and identifies infiltrating monocytes and CD8$^+$ T cells. **d** Violin plot showing the expression of marker genes within the 4 different clusters detected in (C). **e** UMAP clustering of the microglia/monocytes cluster indicating that the expression of the type I interferon signaling pathway (GO:0060337) is particularly obvious in the microglia from *Rnaset2*$^{-/-}$ mice and within the infiltrating monocytes and CD8$^+$ T cells. **f** Columns 1–4: IHC for neurons (NeuN), T cells (CD3), macrophages/activated microglia (MAC3) and ISG15 in the cortex of *Rnaset2*$^{-/-}$ (representative of $n = 5$; upper panel) and control (representative of $n = 6$; lower panel) mice demonstrates the diffuse and perivascular infiltration of T cells (CD3) and macrophages/activated microglia (MAC3) and an extensive expression of ISG15 in *Rnaset2*$^{-/-}$ mice. Scale bar, 50 μM. Columns 5–10: Exemplary co-immunofluorescence staining confirm expression of ISG15 in astrocytes (GFAP), microglia/macrophages (IBA1), and endothelial cells (CD31) in *Rnaset2*$^{-/-}$ mice. Scale bar, 10 μM.

GAAGTGGG-3'), P2 (5'-CGCTTTTCTGGATTCA-TCGACTGTGG-3'), and P3 (5'-GACCCAGGGGCTCAGTACTAGTG-3') producing a product of 838 bp for WT *Rnaset2a* allele and 870 bp for the transgene. For *Rnaset2b*-HDR allele genotyping was performed by genomic PCR with primers P4 (5'-CTGTCCTCTGT GGTGTGTATG-3') and P5 (5'-TGGAAGTATACCC-TCTCTCGGC-3') followed by Sanger sequencing with the same primers.

***Rnaset2*$^{-/-}$ *Ifnar1*$^{-/-}$ double-knockout mice.** For the *Rnaset2*$^{-/-}$ *Ifnar1*$^{-/-}$ double-knockout mice, homozygous *Rnaset2*$^{-/-}$ mice were crossbred with homozygous B6.129S2-*Ifnar1*$^{tm1Agt}$/Mmjax (*Ifnar1*$^{-/-}$) kindly provided by Prof. Brück, Institute of Neuropathology, University Medical Center Göttingen[59]. Routine genotyping of the *Ifnar1*$^{-/-}$ knockout allele was performed by PCR with primer 1 (5'-CGAGGCGAAGTGGTTAAAAG-3'), primer 2 (5'-ACGGATCAAC CTCATTCCAC-3') and primer 3 (5'-AATTCGCCAATGACAAGACG-3') producing a product of 155 bp for wild type Ifnar1 allele and 250 bp for the mutant allele.

**Breeding and maintenance.** Mice were bred and kept in the animal facility of the Max Planck Institute of Experimental Medicine (Göttingen, Germany) and in the facility of the University Medical Center Göttingen, Georg August University (Göttingen, Germany) with 21 °C ambient temperature, 60–65 % humidity, a 12 h light/dark cycle and 2–5 mice per cage. All experiments were performed in accordance with the German animal protection law and with the permission of the Lower Saxony Federal State Office for Consumer Protection and Food Safety (LAVES) under the protocol No. 17-2697.

**Protein extraction and western blotting.** Heterozygous control and *Rnaset2*$^{-/-}$ mouse embryonic fibroblasts (MEFs) were prepared according to the protocol of Durkin et al. and maintained as primary cell cultures[60]. Briefly, day 13 embryos were harvested from the uterus of a female mouse, washed in PBS and the head and the red tissue (heart and liver) removed. The rest of the embryos were minced with scissor and scalpel and finally digested for up to 20 min. at 37 °C in 0.25% trypsin-EDTA. The suspension was transferred to a 50 ml tube and let sit for about 5 min. Finally, the supernatant from three embryos was transferred to a T75 cell culture flask in MEF culture medium (Dulbecco's Modified Eagle's Medium (DMEM) containing 4.5 g/L D-glucose (Life Technologies, Gibco®) supplemented with 10% fetal bovine serum, 1×200 mM L-glutamine and 1x penicillin-streptomycin). The medium was exchange the next day. After a maximum of two passages, cells were harvested and lysed on ice in RIPA buffer containing Complete Protease and Phosphatase Inhibitor Cocktail (Roche). Protein concentration was determined by using the Interchim BC Assay Kit (Interchim) and bovine serum albumin was used as standard. Protein lysates were denatured by boiling in Tris-Glycine SDS sample buffer, separated by SDS-PAGE using 4–20 % Tris-Glycine Mini Gels (Thermo Fisher Scientific) and transferred onto 0.45 μm nitrocellulose membranes (Merck). Subsequently, nitrocellulose membranes were blocked with 5 % non-fat milk in TBS supplemented with 0.1 % Tween 20 (TBST) for one hour at room temperature. Next, primary antibodies were applied in 5 % non-fat milk in TBST overnight at 4 °C (anti-RNaseT2 (1:500), Cloud-Clone Corp., Katy, USA; anti-β-actin (1:15.000), Sigma-Aldrich). On the next day, membranes were washed with TBST and each primary antibody was followed by incubation with either goat anti-rabbit or donkey anti-mouse secondary antibodies conjugated to HRP (Dianova), diluted 1:5000 in 5 % non-fat milk in TBST for one hour at room temperature. Finally, chemiluminescent signal was revealed by using the Lumi-Light Western Blotting Substrate (Merck) and an ImageQuant LAS 4000 Mini imaging system (GE Healthcare). Equal loading of protein lysates was determined by β-actin Western blotting.

**Histology and immunohistochemistry.** For histology, immunohistochemistry (IHC) and organ weights, mixed-sex mice between the age of 4 and 10 months were anesthetized with ketamine (250 mg/kg i.p., Ursotamin® (Ketamin),

Serumwerk) and medetomidine (2 mg/kg i.p., Cepetor, CP-Pharma). Immediately after cardiac arrest, the animals were transcardially perfused with phosphate-buffered saline (PBS, pH 7.4) followed by 4 % paraformaldehyde in PBS (pH 7.4). Organs were dissected out after perfusion and weighted on an analytical balance (2001 MP2, Sartorius). Organ specimens (liver, spleen, kidney, bone, and brain) were post-fixed in 4 % paraformaldehyde in PBS for at least 72 h at 4 °C, embedded in paraffin and cut in three to five μm sections.

First, detailed organ morphology was assessed by hematoxylin-eosin (HE) staining. For IHC, organ sections were de-paraffinized and steam-cooked in antigen target retrieval solution pH 9 (Agilent Dako) according to the manufacturer's protocol. After washing the sections with Tris-buffered saline (TBS) supplemented with 0.025% Triton™ X-100 (TBSTri), blocking was performed with 10% FCS with 1% BSA in TBS for 30 min at room temperature. Next, sections were incubated with primary antibodies: anti-CD41 (1:100, EPR4330, Abcam), rabbit anti-CD3 (1:150, mAb SP7, DCS), rat anti-Mac3 (1:200, mAb M3/84, BD Pharmingen) and anti-B220 (1:100, RA3–6B2, BD Pharmingen) in 1% BSA in TBS overnight at 4 °C. After washing the sections with TBSTri on the next day, endogenous peroxidase was inactivated by 0.3% H$_2$O$_2$ incubation for 17 min at room temperature. Next, sections were incubated with horseradish peroxidase (HRP) conjugated goat anti-rabbit secondary antibody (1:300) (Dianova) for one hour at room temperature. The immunoreactivity was visualized using the SignalStain DAB Substrate Kit (Cell Signaling) by employing DAB as chromogen. Finally, sections were counterstained with Mayer's hemalum (Merck). Images were taken with an Axio Imager.M1 microscope (Zeiss) using 10×/0.3, 20×/0.5 and 40×/ 0.75 Plan-NEOFLUAR objectives and an Axiocam 105 color camera (Zeiss).

For the analysis of CNS tissue, histological staining comprised hematoxylin-eosin (HE), Luxol fast blue–periodic acid Schiff (LFB-PAS) and Bielschowsky silver impregnation. For immunohistochemistry, the following primary antibodies were used: mouse anti-glial fibrillary acidic protein (GFAP, 1:300, 173 011, Synaptic Systems), mouse anti-ionized calcium binding adapter molecule 1 (Iba1, 1:100, MABN92, Merck/Millipore), rabbit anti-NeuN (NeuN, 1:3000, ab177487, Abcam), rabbit anti-transmembrane protein 119 (TMEM119, 1:100, ab209064, Abcam), rabbit anti-interferon stimulating gene 15 (ISG15, 1:100, kindly provided by Marco Prinz), mouse anti-Platelet endothelial cell adhesion molecule (CD31, 1:10, DIA310M, Dianova), rat anti-myelin basic protein (MBP, 1:100, mAb12, Abcam), rabbit anti-CD3 (1:150, mAb SP7, DCS), rat anti-Mac3 (1:200, mAb M3/84, BD Pharmingen) and rat anti CD8a (1:100, 4SM15, Invitrogen). Antibody-binding was detected using a peroxidase/DAB protocol, as described above. For CD3-CD8 double immunohistochemistry, fluorescently labelled secondary antibodies were applied. Sections were evaluated using an Olympus BX51 light or combined light and fluorescent microscope. To analyze ISG15 expression on whole slide images of whole hemispheres color deconvolution (scikit-image, v0.18.1) of the DAB-ISG15 signal was performed (Supplementary Fig. 13)[61].

**Blood, urine, and bone marrow analysis.** For the collection of blood samples mixed-sex mice between the age of 4 and 10 months were deeply terminal anesthetized with ketamine (250 mg/kg i.p.) and medetomidine (2 mg/kg i.p.) for cardiac puncture.

To obtain serum, whole blood samples were stored at room temperature for one to two hours before being separated by centrifugation and stored at −20 °C. In order to measure clinical chemistry parameters, serum samples of 11 *Rnaset2*$^{-/-}$ and 13–16 control mice were analyzed in the accredited interdisciplinary laboratory of the Institute of Clinical Chemistry of the University Medical Center Göttingen by standard protocols.

Peripheral blood counts were analyzed from ethylenediaminetetraacetic acid (EDTA)-anticoagulated whole blood samples in a threefold dilution using a Microsemi CRP Analyzer (Horiba) within four hours after blood withdrawal. Knockout and control animals analyzed $n = 13$–20, respectively.

For counting reticulocytes, EDTA-stabilized blood samples from each mouse were stained with brilliant cresyl blue (KABE REF017005) according to the manufacturer's recommendation, scratched out and analyzed by the "meander"

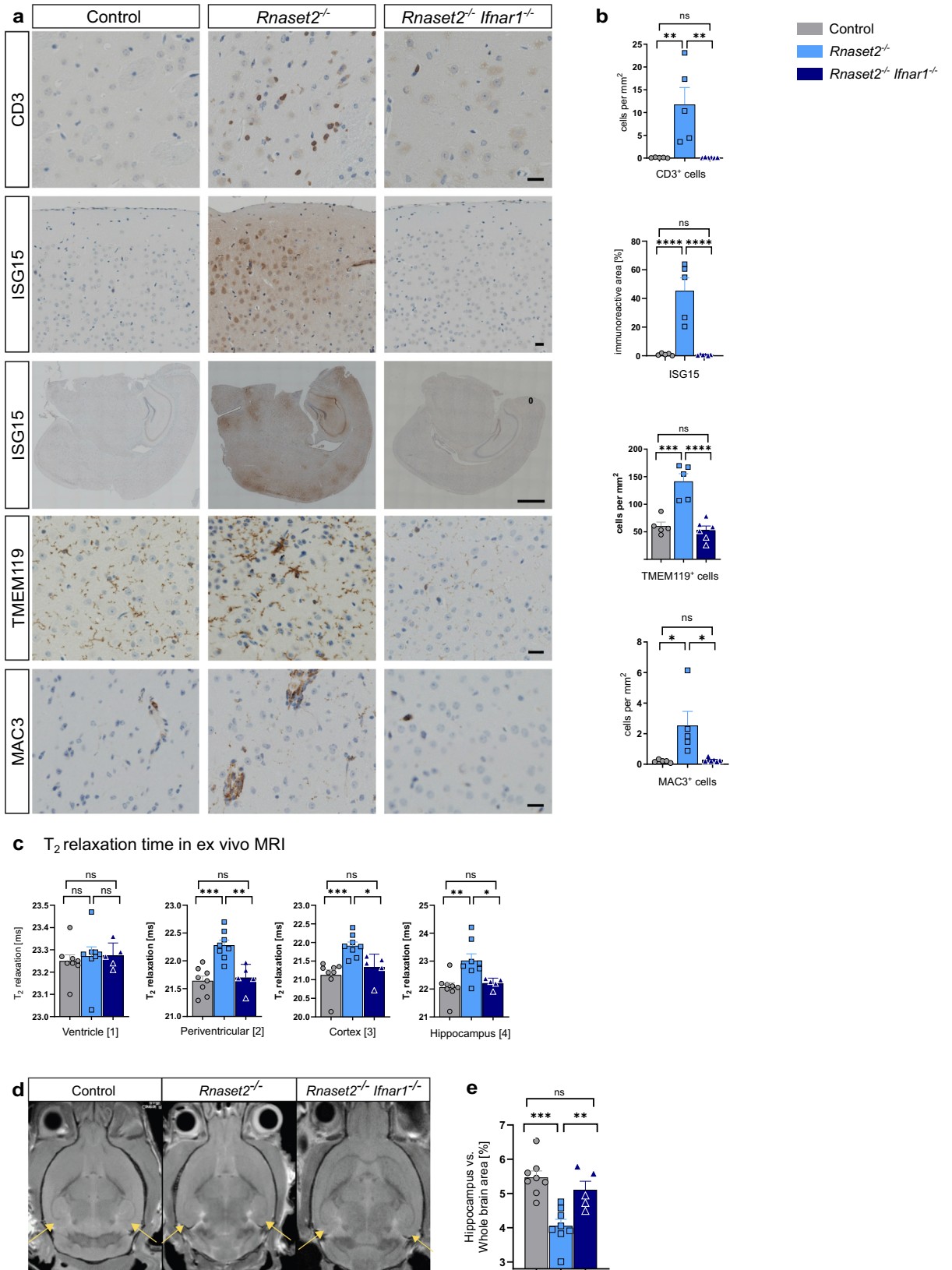

technique. At least 1000 erythrocytes were counted for each slide at a magnification of 1000-fold to determine the number of reticulocytes.

Urine samples from *Rnaset2*−/− and control mice were collected during routine handling. In general proteins from urine samples are extracted by chloroform-methanol precipitation. Briefly, 20 µl of urine was diluted with water to a final volume of 100 µl. Addition of 4 volumes methanol, 1 volume chloroform and 3 volumes water leads to extraction of total protein from the urine sample. After

centrifugation, the supernatant was dried and proteins were denatured by boiling in Tris-Glycine SDS sample buffer and separated by SDS-PAGE using 4–20% Tris-Glycine Gel (NovexTM, Thermo Fisher Scientific). 300 ng bovine serum albumin (BSA) was loaded as standard control and 10 µl molecular marker (Pro Sieve Quad Proteinmarker 4.6–300 kDa, Biozym Scientific) was used.

The bone marrow from *Rnaset2*−/− and control mice was isolated after cardiac arrest from the femur. In brief, the femur was segregated from the mouse by

**Fig. 8 *Ifnar1*$^{-/-}$ knockout rescues *Rnaset2*$^{-/-}$ related neuropathology. a** Immunohistochemistry of rescued cerebral T cell infiltration (CD3), interferon-stimulated gene 15 (ISG15) upregulation, microglial activation (TMEM119), and macrophages/activated microglia infiltration (MAC3) in *Rnaset2*$^{-/-}$ *Ifnar1*$^{-/-}$ mouse brain sections compared to control mouse and *Rnaset2*$^{-/-}$ mouse brain sections. Scale bar, 20 μM and 1 mm for whole slices of ISG15. **b** Rescued cerebral inflammatory infiltration of CD3$^+$ and MAC3$^+$ cell densities determined in whole brain slices, TMEM119$^+$ cell density of representative cortical regions, and % of immunoreactive area for ISG15 in *Rnaset2*$^{-/-}$*Ifnar1*$^{-/-}$ ($n = 6$) compared to control ($n = 5$) and *Rnaset2*$^{-/-}$ ($n = 5$). Images in **a** are representative for the 5–6 animals analyzed for each genotype. **c** T$_2$ relaxation time measurement in various brain regions of *Rnaset2*$^{-/-}$*Ifnar1*$^{-/-}$ ($n = 5$), compared to control ($n = 8$) and *Rnaset2*$^{-/-}$ ($n = 8$). For regions of interest (ROI) 1–4 selected for T$_2$ relaxation time measurements see Fig. 5d. IFNAR1-deficiency resolved extension of T$_2$ relaxation in *Rnaset2*$^{-/-}$ mice. Data of two independent experiments. **d** Representative magnetic resonance anatomical images in axial plane showing ventricular enlargement in *Rnaset2*$^{-/-}$ mice but not in *Rnaset2*$^{-/-}$*Ifnar1*$^{-/-}$ mice due to shrinkage of the hippocampus on both sides (arrows). **e** Percent area of hippocampus (left and right) with respect to whole brain area of *Rnaset2*$^{-/-}$ ($n = 8$), control ($n = 8$) and *Rnaset2*$^{-/-}$ *Ifnar1*$^{-/-}$ ($n = 6$). Data depicted as the mean ± SEM. *p* values of one-way ANOVA are represented as ****$p < 0.0001$, ***$p < 0.001$, **$p < 0.01$, *$p < 0.05$ and not significant (ns) $p \geq 0.05$. MRI data from *Rnaset2*$^{-/-}$ and control mice are also depicted in Fig. 5c and d. Source data are provided as a Source data file.

cutting underneath the hip bone and above the knee. The tissue surrounding the femur was segregated, and the bone marrow was pushed out of the femur with the help of a bone marrow puncture needle. The bone marrow was placed on a microscopic slide, mixed up with 30 μl physiological saline and scratched out. For detailed analysis, the bone marrow smears were stained with Giemsa stain in the accredited laboratory of the Institute of Pathology of the University Medical Center Göttingen by standard protocols.

**FACS analysis and antibodies**. Single-cell suspensions were prepared by mashing spleen mixed-sex mice between the age of 4 and 10 months through a 70 μm cell strainer (Greiner Bio-One). Cells were washed with FACS buffer (PBS + 2% FBS) and incubated for 15 min in Fc-blocking buffer and stained with the following anti-mouse antibodies (if not stated otherwise, all from BioLegend or eBioscience) for 30 min on ice protected from light: B220 (RA3–6B2), c-kit (2B8), CD34 (HM34), CD41 (MWReg30), CD44 (IM7), CD45 (30F11), CD71 (RI7217), CD105 (MJ7/18), CD150 (TC15-12F12.2), Gr1 (RB6-8C5), SCA1 (D7), Ter119 (TER-119), CD16/32 (93). 7-aminoactinomycin D (7-AAD, Thermo Fisher Scientific) was used to assess viability. Cells were washed twice and resuspended in FACS buffer, before flow cytometry analysis was performed using a BD LSRFortessa$^{TM}$ X-20 (BD Biosciences). Data were collected by BD FACSDiva Software v8.0.2 (BD Biosciences) and analyzed using the FlowJo$^{TM}$ Software Version 10.6.1 (Becton, Dickinson and Company, 2019).

Hematopoietic fractions were determined according to the following immunophenotypes: Erythrocytes all B220− and Ter119+: CD44− (R5), CD44+ low cellular volume (R4), CD44+ medium cellular volume (R3), CD44+ high cellular volume (R2). B220− Ter119− Gr1− (lineage−) c-Kit+Sca1− (LKS−) CD34−CD16/32−, megakaryocyte–erythroid progenitor (MEP); LKS −CD34+CD16/32−, common myeloid progenitor (CMP); LKS−CD34+CD16/32+, granulocyte–macrophage progenitor (GMP); LKS−CD150+CD41+, megakaryocyte progenitor (MkP); LKS−CD41−CD16/32−CD150−CD105−, Pre-GM; LKS−CD41−CD16/32−CD150+CD105−, pre-megakaryocyte-erythroid (Pre-Meg-E); LKS−CD41−CD16/32−CD150+CD105+, pre-colony-forming unit erythroid (Pre-CFU-E); LKS−CD41−CD16/ 32−CD150−CD105+, CFUE.

**Preparation of CNS mononuclear cells**. Mixed-sex mice were transcardially perfused with 20 ml PBS at the age of 4–10 months. Brain and spinal cord were removed and dissociated with the gentle MACS$^{TM}$ dissociator. The tissue was digested for 35 min with 2.5 mg/ml collagenase D (Roche) and 1 mg/ml DNaseI (Roche) at 37 °C. Mononuclear cells were isolated by Percoll gradient centrifugation (37%/70%, GE Healthcare) removed from the interphase, washed and subsequently blocked with αCD16/32 (BioLegend, Clone 93) for 15 min. The following antibodies were used (all from BioLegend or eBioscience): αCD3 (145-2C11), αCD4 (RM4-5), αCD8 (53-6.7), αCD11b (M1/70), αCD19 (1D3), αB220 (RA3-6B2), αTCRβ (H57-597), αγδ TCR (GL3), αCD44 (IM7), αCD62L (MEL-14), αCD45 (30-F11), αLy6C (HK1.4), αLy6G (1A8), αCCR2 (R&D 475301), αP2RY12 (S16007D), αH2Kb/Db antibody (28-8-6), mouse IgG2a isotype control (MOPC-173). Flow cytometry analysis was performed using a BD LSRFortessa$^{TM}$ X-20 (BD Biosciences, CA) or a FACS Canto$^{TM}$ II device (BD Biosciences, CA). Results were analyzed using FlowJo$^{TM}$ Software Version 10.6.1 (Becton, Dickinson and Company, 2019). Naïve CD8 T cells were defined as CD45$^+$ TCRβ$^+$ CD8$^+$ CD44$^{low}$ CD62L$^+$, effector memory CD8 T cells as CD45$^+$ TCRβ$^+$ CD8$^+$ CD44$^{hi}$ CD62L$^-$ and central memory CD8 T cells as CD45$^+$ TCRβ$^+$ CD8$^+$ CD44$^{hi}$ CD62L$^+$. Inflammatory monocytes were defined as CD45$^+$ CD11b$^+$ Ly6C$^{hi}$ CCR2$^+$ cells. Microglia cells for MFI assessment were defined as CD45$^{int}$ CD11b$^+$ Ly6C$^-$ Ly6G$^-$ P2RY12$^+$.

**RNA extraction, reverse transcription, and quantitative real-time PCR (qPCR)**. A total RNA of six mixed-sex control and six *Rnaset2*$^{-/-}$ mice between 4 and 8 months were used for RNA extraction from the liver, spleen and kidney using FastGene RNA Premium Kit (Nippon Genetics) and from the brain using

peqGOLD TriFast$^{TM}$ (peqlab) according to the manufacturer's instructions. Organs were isolated from *Rnaset2*$^{-/-}$ and control mice after perfusion with PBS (pH 7.5). RNA quality was verified by gel electrophoresis and optical density (OD) measurement. For first-strand cDNA synthesis, 2 μg of RNA was reverse transcribed using SuperScript III First-Strand Synthesis System (Invitrogen) according to the manufacturer's recommendation.

Quantitative PCR for *Ifi27, Ifi44, Ifit1, Isg15, Rsad2, Siglec1, Usp18, Cxcl10, Irf7, Ifih1, Ddx58* and *Rplp0* and *Tbp* (as internal controls) were performed using PrimeTime qPCR Assays listed in Supplementary Table 2 and ordered by Integrated DNA Technology (IDT). All reactions were done in triplicates on a QuantStudio 3 Real-Time PCR System (Applied Biosystems) according to the manufacturer´s recommendations in reactions containing 20 ng cDNA. The relative gene expression values were quantified according to the $\Delta\Delta_{CT}$ method[62] by QuantStudio$^{TM}$ Design & Analysis Software v1.4.3, and the values of controls were normalized to 1.

**Detection of anti-nuclear antibodies**. The standard method for detecting anti-nuclear antibodies (ANA) is the indirect immunofluorescence test (IIFT). ANA measurements were performed by EUROIMMUN AG (Lübeck, Germany) using the HEp-2 IIFT Dog (IgG) test kit and an anti-mouse-IgG-FITC antibody. *Rnaset2*$^{-/-}$ and control mouse sera were tested. In brief, BIOCHIPS coated with HEp-2 cells were incubated with differently diluted mouse sera. In a positive reaction, specific antibodies of the IgG class bind to antigens and in a second step, attached antibodies are stained with fluorescein-labelled anti-mouse antibodies and visualized using a fluorescence microscope. Results are reported as titres and titres are expressed as ratios obtained by dilution of the sample with saline. To evaluate anti-nuclear antibody prevalence, samples were further titrated until a fluorescence signal was just visible.

**Ex vivo magnetic resonance imaging**. For cerebral magnetic resonance imaging (MRI), mixed-sex mice between the age of 4 and 7 month were anesthetized with ketamine (250 mg/kg i.p.) and medetomidine (2 mg/kg i.p.). Immediately after cardiac arrest, the animals were transcardially perfused with PBS (pH 7.4) followed by 4 % paraformaldehyde in PBS (pH 7.4). Mice heads were separated and post-fixed in 4% paraformaldehyde in PBS (pH 7.4) for at least one week at 4 °C. Before MRI measurements, the head were embedded in Fomblin (perfluoropolyether). All MRI measurements were conducted on a 300 MHz (7 T) vertical wide bore system, using a transmit/receive birdcage radiofrequency coil with an inner diameter of 25 mm and a 1 Tm$^{-1}$ gradient insert (Bruker). The system was interfaced to a Linux operating system running Topspin 2.0 and Para Vision 3.0 imaging software (Bruker Biospin). Before each measurement, the magnetic field homogeneity was optimized by shimming. Each session of measurements started with a multi-slice orthogonal gradient-echo sequence for position determination and selection of the desired region for subsequent experiments. T$_2$-weighted MR images were acquired by a rapid acquisition with relaxation enhancement (RARE) sequence, which employs a single excitation step followed by the collection of multiple phase-encoded echoes[63]. Basic measurement parameters used for the RARE sequence were as follows: effective echo time = 18.1 ms; repetition time = 5 s; flip angle = 90°; averages = 32 (for sagittal and axial) or 64 (for coronal) and RARE factor (echo train length) = 4; image slice thickness = 0.5 mm. The field of view was 2.5 × 2.5 cm$^2$, with an image matrix of 256 × 256. This yields an effective in-plane resolution of ~97 μm. For measurement of brain areas, the desired region of interest was drawn on the image and areas were computed using an image sequence analysis (ISA) tool package (Paravision 5, Bruker). The data were exported to OriginPro v. 8 (OriginLab) for further analysis and percentage of specific area with respect to whole brain area was calculated. One-way ANOVA (Tukey's post-test) for comparison of mean between groups was performed. Levene's test was performed for homogeneity of variance analysis.

For transverse relaxation time (T$_2$) measurement, a standard multi-slice multi-echo (MSME) sequence was used. This sequence is based on the Carr-Purcell Meiboom-Gill (CPMG) sequence, where transverse magnetization of a 90° pulse is

refocused by a train of 180° pulses generating a series of echoes. The following imaging parameters were used: nominal flip angles = 90° and 180°, and a train of 12 echoes with TEs ranging from 8.17 ms to 98 ms with 8.17 ms echo-spacing; TR = 2 s, slice thickness = 0.5 mm; number of slices 8 and a matrix size 256 × 256 pixels. For calculation of $T_2$ relaxation time, regions of interest (ROIs) were drawn at desired locations within the brain using an image sequence analysis (ISA) tool package (Paravision 5, Bruker). Monoexponential fitting was then used to calculate $T_2$ using a monoexponential fit function [y = A + C*exp (–t/$T_2$)], where A = Absolute bias, C = signal intensity, $T_2$ = transverse relaxation time. Means and standard deviation for $T_2$ relaxation times for each ROI were calculated.

**In vivo magnetic resonance imaging**. For this analysis two male $Rnaset2^{-/-}$ and two male control mice at age of 10 months were assessed. Mice were anesthetized with ketamine (60 mg/kg, i.p.) and medetomidine (0.4 mg/Kg, i.p.), subsequently intubated and maintained under anesthesia by active ventilation with isoflurane (0.5–1.75% in ambient air and oxygen (1:1)). MRI data were acquired at a magnetic field strength of 9.4 Tesla (94/30 BioSpec, Bruker BioSpin) using a saddle shaped 4-channel phased array coil for signal detections. Multi-slice T2-weighted images* (RARE, TR/effective TE = 6843/55 ms, number of echoes 8) were obtained with a special resolution of 40 × 40 × 300 μm³. For dynamic contrast enhanced MRI (DCM) a 3D T1-weighted data set (FLASH, TR/TE = 18/3.4, flip angle = 8°, 100 μm isotropic spatial resolution) was acquired before ($S_0$) and 20 min after ($S_C$) intravenous injection (via tail vein catheter) of 0.3 mmol/Kg body weight Gd-DTPA (Magnevist®, Bayer Vital GmbH, diluted to 0.1 mM in physiological saline). Signal enhancement in percentage was calculated as followed: ($S_C$–$S_0$)/$S_0$*100. 3D-reconstruction of the brain and the lateral ventricles was performed based on magnetisation transfer-weighted data sets (3D FLASH, TR/TE = 15.2/3.4 ms, flip angle = 5°, Gaussian pulse with an off-resonance frequency of 3000 Hz and a power of 6 μT, 100 μm isotropic spatial resolution) using the software package Amira 6.2.0 (Thermo Fisher Scientific) for semiautomatic segmentation.

**Behavior testing**. For all behavioral analysis a cohort of 17 $Rnaset2^{-/-}$ and 15 control mice, at an age of 4 months, with mixed sex, was assessed.

The balance beam task was used to assess balance and general motor performance. A wooden beam of 1 cm diameter and a length of 50 cm was attached to two 9 × 15 cm support columns, which were elevated at a height of 44 cm above a padded surface. Mice were positioned on the beam in a central position and released. During a single day of testing, each mouse was given three consecutive 60 s trials with at least 10 min intervals in between. For mice remaining on the beam for the entire trial period or managing to reach one of the escape platforms, the maximum time of 60 s was recorded and an average latency to fall was calculated[64].

Neuromuscular abilities and muscle strength were tested using the inverted grid test. The testing apparatus consisted of a 45 cm × 30 cm wire grid 45 cm wire grid with 1 mm wires and a grid spacing of 1 cm. The grid was suspended 40 cm above a padded surface. Latency to fall was recorded during a single 60 s trial after a mouse had been placed onto the center of the grid and turned upside down. If the mice remained on the grid for the entire trial, the maximum time of 60 s was recorded[65].

The novel object recognition (NOR) test relies on three independent sessions: one habituation session (Day0), one training session (Day1), and one test session (Day2), 5 min each, performed on 3 consecutive days. The open field test was used as habituation phase, thereby assessing both exploratory behavior and locomotor activity. Mice were tested using an open field box made of grey plastic with 50 cm × 50 cm surface area and 38 cm-high walls. Monitoring was accomplished by an automated tracking system (ANY-maze v5.1, Stoelting Co.) and the running speed was recorded during 5-min sessions. The training session for the NOR analysis simply involves visual exploration of two identical objects, while one of these objects is replaced with a novel object in the test session. Because rodents have an innate preference for novelty, a rodent that remembers the familiar object will spend more time exploring the novel object. On day 1, two similar objects were placed in different quadrants of the open field arena, and the mouse could move freely for 5 min. In order to diminish odor cues, the arena and objects were cleaned with 70 % ethanol solution after each test session. On testing day, one of the two objects from training day was replaced by a novel object. Mice were again allowed to explore freely for a period of 5 min. For data analysis, animals with exploration times below 5 s in one period of 5 min were excluded. The recognition performances were quantified using the Discrimination Ratio (DR), measured as the differences between novel ($T_{novel}$) and familiar ($T_{familiar}$) object exploration times in proportion to the animal's total exploration time ($T_{total}$): ($T_{novel}$ - $T_{familiar}$)/$T_{total}$. A DR equivalent to 0 indicates an equal exploration of the novel and familiar objects, while a positive or a negative value points to a preferential exploration of the novel and familiar object[66].

Spontaneous alternation rates were assessed using a cross maze built from grey plastic material which had four arms arranged in a 90° position. Each arm was 30 cm in length, with 8 cm width and a height of 15 cm. During 10 min test sessions, each mouse was randomly placed in one arm and allowed to move freely through the maze. Alternation was defined as successive entries into the four arms in overlapping quadruple sets (e.g., 1, 3, 2, 4 or 2, 3, 4, 1 but not 1, 2, 3, 1). Percent of spontaneous alternation was defined as the ratio of actual (=total alternations) to possible (=total arm entries −3) number of alternations × 100. In order to diminish odor cues, the maze was cleaned with 70% ethanol solution[67].

**Single nuclei RNA sequencing analysis**. For RNA sequencing analysis two mixed-sex $Rnaset2^{-/-}$ and control mice at the age of 8 months were assessed. Mice were deeply anesthetized with ketamine and medetomidine and transcardially perfused with phosphate-buffered saline (PBS, pH 7.4). After cervical dislocation, the whole brain was removed from the skull and the preparation of the hippocampus and caudate putamen was performed in a petri dish filled with ice cold PBS buffer under a binocular. Tissue sample from two hemispheres of different mice with the same genotype were pooled and immediately frozen on liquid nitrogen and stored at −80 °C until further processing. Nuclei from frozen mouse brain tissues were isolated according to previously published protocol with modifications[68]. Briefly, froze tissues were Dounce homogenized in 500 μl EZ prep lysis buffer (Sigma NUC101) supplemented with 1:200 RNAse inhibitor (Promega, N2615) for 45 times in a 1.5 mL Eppendorf tube using micro pestles. The volume was increased with lysis buffer up to 2000 μl and incubated on ice for 5 min. Lysates were centrifuged for 5 min at 500 × g at 4 °C and supernatants were discarded. The pellet was resuspended into 2000 μl lysis buffer and incubated for 5 min on ice. After 5 min centrifugation (500 × g at 4 °C), the resulting pellet was resuspended into 1500 μl nuclei suspension buffer (NSB, 0.5 % BSA, 1:100 Roche protease inhibitor, 1:200 RNAse inhibitor in 1×PBS) and centrifuged again for 5 min (500 × g at 4 °C). The pellet was finally resuspended into 500 μl NSB and stained with 7AAD (Invitrogen, Cat: 00-6993-50). Single nuclei were sorted using BD FACS Aria III sorter. Sorted nuclei were counted in Countess II FL Automated Counter. ~5000 single nuclei per sample were used for GEM generation, barcoding, and cDNA libraries according to 10× Chromium Next GEM Single Cell 3′ Reagent v3.1 protocol. Pooled libraries were sequenced in Illumina NextSeq 550 in order to achieve >50,000 reads/nuclei.

Gene counts were obtained by aligning reads to the mm10 genome (NCBI:GCA_000001635.8) (GRCm38.p6) using CellRanger software v.4.0.0 (10XGenomics). The CellRanger count pipeline was used to generate a gene-count matrix by mapping reads to the pre-mRNA as reference to account for unspliced nuclear transcripts.

The SCANPY package v1.7.2 was used for pre-filtering, normalization and clustering[68]. Initially, cells that reflected low-quality cells (either too many or too few reads, cells isolated almost exclusively, cells expressing less than 10% of house-keeping genes) were excluded[69]. Next, counts were scaled by the total library size multiplied by 10.000 and transformed into log space. Highly variable genes were identified based on dispersion and mean, the technical influence of the total number of counts was regressed out, and the values were rescaled. Differential expression analysis was performed using the rank_genes_groups function as implemented in SCANPY.

Principal component analysis (PCA) was performed on the variable genes, and UMAP was run on the top 50 principal components (PCs)[70]. The top 50 PCs were used to build a k-nearest-neighbours cell–cell graph with k = 100 neighbours. Subsequently, spectral decomposition over the graph was performed with 50 components, and the Leiden graph-clustering algorithm was applied to identify cell clusters. We confirmed that the number of PCs captures almost all the variance of the data. For each cluster, we assigned a cell-type label using manual evaluation of gene expression for sets of known marker genes. Violin plots for marker genes were created using the stacked_violin function as implemented in SCANPY. GO-term analysis was performed using ShinyGOv0.60[71]. Data files of all differentially expressed genes and GO-term analysis are accessible at Supplementary Data 1.

**Statistics and reproducibility**. Statistical analysis was performed with GraphPad Prism version 8.3.0 (GraphPad Software Inc, San Diego, CA) using two-tailed Student's t-test, two-tailed Welsh's test, one-way ANOVA and two-way ANOVA followed by Tukey´s multiple comparison test unless indicated otherwise. For all tests, p-values less than 0.05 were considered statistically significant. Asterisks indicate the magnitude of the statistical significance: *$p < 0.05$, **$p < 0.01$, ***$p < 0.001$, ****$p < 0.0001$. ns (not significant) $p \geq 0.05$. Error bars indicate the mean with standard error of the mean (SEM).

Survival curves are computed using the open source python package lifelines (version 0.26.0, doi:10.5281/zenodo.4816284) and confidence intervals are estimated using the implemented exponential Greenwood method. The alpha level was corrected using Bonferroni's method by the number of comparisons (alpha = 0.05/3).

Immunofluorescence recordings were partially adjusted in brightness and contrast to different degrees, rotated and cropped using the OMERO.figure add-on (https://www.openmicroscopy.org/omero/figure/). Especially in Fig. 7f GFAP, Iba-1 and CD31 were adjusted for optimal morphology visualization while ISG15 was recorded and displayed using exactly the same conditions for control and $Rnaset2^{-/-}$.

**Reporting summary**. Further information on research design is available in the Nature Research Reporting Summary linked to this article.

## Data availability
Raw data for all results/figures presented in this article, including Supplementary Figures, are available in the Source data file. Any additional data are available from the corresponding author upon reasonable request. Raw single nuclei RNA sequencing data are accessible from the NCBI GEO database using the accession number GSE180138.

Detailed annotation tables for these datasets are provided in Supplementary Data 1. Mus musculus reference genome (GRCm38.p6) use for single RNA nuclei study is accessible from NCBI GenBank under the accession numbers GCA_000001635.26. Reference sequence for *Rnaset2a* and *Rnaset2b* gene are accessible from NCBI GenBank under the accession numbers NM_001083938.3 and NM_026611.2, respectively. Source data are provided with this paper.

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

## Acknowledgements

MK was supported by the Göttinger College for Translational Medicine by the Lower Saxony Ministry of Science and Culture. A.A.i. was supported by the European Research Council (ERC) Advanced Grant (ERCAdG No. 339580). M.S.S., A.F., P.R., A.A.i. and J.G. were funded by the Deutsche Forschungsgemeinschaft (DFG, German Research Foundation) under Germany's Excellence Strategy—EXC 2067/1-390729940, E.B. and G.H .by EXC2151—390873048 and S.N., C.St., and J.G. by TRR274. A.F. received funds from the ERC consolidator grant DEPICODE (648898), the BMBF project ENERGI (01GQ1421A), the DFG SFB1286 and the Hans and Isle Breuer foundation. A.F. and J.G. had funds from the German Center for Neurodegenerative Diseases. S.N. (NE 2447/1-1), C.St. (STA1389/5-1) and J.G. (Ga354/16-1) were funded by DFG research grants. The authors thank the transgenic core facility of the Max Planck Institute of Experimental Medicine and especially Ursula Fünfschilling for microinjections and helpful discussions in generating the *Rnaset2⁻/⁻* mouse strain. We thank EUROIMMUN AG Lübeck, Germany for measuring of anti-nuclear antibodies (ANA), Hans-Joachim Anders for providing a positive control serum for ANA detection and Oliver Gross for urine protein analysis as well as Marc Ziegenbein, Elisabeth Ehbrecht, Susanne Burkhardt, Heidi Brodmerkel, and Olga Kowatsch for excellent technical assistance.

## Author contributions

M.K., M.H., S.N. and J.G. designed and supervised the project. M.K., K.T., and K.W. performed and analyzed most experiments. S.Z. and O.W. performed the behavior testing in mice. C.Sch. performed the qPCR experiments for interferon-stimulated genes. J.F., A.W., C.St. and S.N. designed and analyzed the neuropathology experiments. S.N. supported and performed part of the FACS analysis. A.Ai. and P.R. supported the CRISPR–Cas9 knockout mouse generation. K.W., S.H. and J.K. performed and analyzed histopathology, blood smear and bone marrow analysis. H.W. and S.S. supported the generation of RNT2A knockout mouse. S.B. and A.Al. performed and analyzed the MRI experiments. RE prepared tissue samples for single nucleus sequencing. M.S.S. isolated nuclei. M.S.S. and L.K. prepared sequencing libraries and performed next-generation sequencing. D.M.K. performed quality control, mapped and analyzed single nucleus sequencing data. D.M.K. and A.F. interpreted single nucleus sequencing data and A.F. generated the figures. M.K., K.T., S.N. and J.G. wrote the manuscript supported by K.W., M.P., E.B., G.H., M.H., O.W., A.F. and C.St. All authors read and approved the manuscript.

## Competing interests

The authors declare no competing interests.
