## [Peer Review File · Nature Communications]

Interferon-driven brain phenotype in a mouse model of RNaseT2 deficient leukoencephalopathyREVIEWER COMMENTS

Reviewer #1 (Remarks to the Author):

This is an interesting and nicely written manuscript reporting a newly generated mouse model of RNaseT2 deficiency. The model is novel (in mouse), although there are models of RNASET2 deficiency in zebrafish and rat. The paper is quite simple, and essentially descriptive.

In my opinion, the authors do two things that are of potential importance: 1. They convincingly demonstrate that their mouse model demonstrates an upregulation of type I interferon signalling; 2. They show that their mouse model has a neurological phenotype.

In all honesty I don't have much concern about the findings presented in the manuscript. My disappointment comes because the authors really could go further with this – to define the pathway mediating the upregulated signalling in their model, and thereby prove, or not, that the neuropathology they report is definitely related to a dysregulation of interferon. This is where the prize lies – the situation as to the function of RNASET2 is confusing, with previous papers suggesting that it is necessary for the processing of RNA species so as to signal to TLR8 - in which case a failure of enzymatic activity would not obviously induce interferon signaling. The molecule has also been suggested to have a role in the processing ribosomal RNA, and in mitochondrial function.

A few thoughts:

The authors state that their model recapitulates "...the devastating, interferon-induced neuroinflammatory encephalopathy found in patients". I think that this is going a bit far – in is not clear that the neurological phenotype in RNASET2 deficiency in humans is interferon driven. It might be, but there is not so much evidence in support of that possibility. I think that they should tone this statement down. This issue relates to my point above – the authors could use their mouse to provide important evidence as to the basis of the neuropathology of the disease (at least in their mouse) – I don't quite understand why they are publishing now. Have the authors crossed their mouse with an IFNAR-null mouse? Until they do that, they cannot claim that the neurological phenotype is interferon induced.

Have the authors crossed their mouse with a MAVS or STING-null mouse? I think that they should be more agnostic about the mechanism until the relevant crosses have been done. For example, perhaps RNASET2 has a role in the ER-Golgi axis on STING signalling. As such, it seems to me that their Figure 8 is pretty much pure speculation.

Is there any evidence of rRNA accumulation (their reference 16)? What about mitochondria (their reference 19)?

The mice appear to demonstrate a lethal systemic inflammation: is there any evidence of antibody production and immunological kidney disease (a murine 'lupus' like phenotype)?

Reviewer #2 (Remarks to the Author):

This study describes a new experimental model of RNaseT2 deficiency in mice, which leads to a generalized type1 interferonopathy, which also is associated with CNS neurodegeneration. The topic is interesting, since the induction of a type 1 interferonopathy, which also affects the central nervous system, in a mouse model provides an interesting tool to study not only Aicardi-Goutieres Syndrome (AGS), but may also elucidate basic mechanisms of virus induced neurodegeneration. The study describes in detail the model, the clinical and pathological phenotype and provides clear evidence that RNaseT2 deficiency induces an interferonopathy. Further studies on the pathogenetic mechanisms of neurodegeneration are not provided.

Critique:

A major problem of the study is that it is not fully clear, what is the aim of the study.

It could be the description of the new model and a detailed discussion of its usefulness, its advantages over other models and its limitations. In this respect the study is rather incomplete.

Modeling of human RNaseT2 deficiency is rather imperfect. The human disease is predominately a leukodystrophy. Whether there are white matter lesions in the respective model and to what extent they reflect the situation in humans is not addressed in this study. How the grey matter lesions, described in this study, really compare to those in patients is also not described and discussed in depth. In this respect a table comparing the clinical and pathological phenotype of this model with the human disease and the AGS models would be important. Furthermore, it would be important to differentiate, how much of the described phenotype is a direct consequence of RNaseT2 deficiency, and what is really triggered by the interferonopathy.

Another aim could be to use such a new model to unravel the molecular mechanisms of neurodegeneration. This has been done in part before in another model of Type 1 interferon driven neurodegeneration, which has previously been published in Nature Communications (Rubino et al). For such a study one would have to identify the cells in the CNS, which drive the lesions, to separate type 1 interferon effects from RNaseT2 deficiency alone, to define the cellular source of type 1 interferons in the CNS and what cells are affected by type 1 IFN signaling. The authors put much emphasis on the infiltration of the CNS by MHC Class I restricted T-cells, but whether these cells are just secondarily recruited or play a role in the neurodegenerative disease remains unresolved. Also the differential involvement of microglia versus recruited macrophages and their role in neurodegeneration has not been elucidated. In this respect the authors have included Figure 8, depicting a potential pathomechanistic model, but the study does not contain the respective supportive data.

Brief overview of additional experiments performed:

1. Comparative single nuclei RNA sequencing analysis of KO and WT brain
2. Immunohistochemical staining of ISG15 in different CNS cells
3. Organ size, blood counts and neuropathological sections of the newly generated *Rnaset2^{-/-}Ifnar1^{-/-}* mouse
4. Indirect immunofluorescence of anti-nuclear antibodies from serum of KO and WT mice
5. Histological images from kidney and urine protein analysis

Reviewer #1: This is an interesting and nicely written manuscript reporting a newly generated mouse model of RNaseT2 deficiency. The model is novel (in mouse), although there are models of RNASET2 deficiency in zebrafish and rat. The paper is quite simple, and essentially descriptive.

In my opinion, the authors do two things that are of potential importance: 1. They convincingly demonstrate that their mouse model demonstrates an upregulation of type I interferon signalling; 2. They show that their mouse model has a neurological phenotype.

In all honesty I don't have much concern about the findings presented in the manuscript. My disappointment comes because the authors really could go further with this – to define the pathway mediating the upregulated signalling in their model, and thereby prove, or not, that the neuropathology they report is definitely related to a dysregulation of interferon. This is where the prize lies.

In order to address this important point, we have included very recent data on *Rnaset2^{-/-}Ifnar1^{-/-}* mice, which we have crossbred for this study. Here, the neuropathological phenotype of *Rnaset2^{-/-}* mice is clearly improved with concurrent IFNAR deficiency. (Please see new figure 8.) In addition, organ enlargement and hematopoiesis also improved in *Rnaset2^{-/-}Ifnar1^{-/-}* mice.

Moreover, we have also added comprehensive data from single nuclei RNA sequencing analysis providing import insights to the interferon signaling in different CNS cell types and mechanistic insights to the observed neuroinflammation. (Please see new Figure 7 and Supplemental Figure 7.)

The situation as to the function of RNASET2 is confusing, with previous papers suggesting that it is necessary for the processing of RNA species so as to signal to TLR8 - in which case a failure of enzymatic activity would not obviously induce interferon signaling.

We absolutely agree that the immunological function of RNASET2 is confusing. It is certainly correct that a reduced excitation of a TLR does not lead to an induction of interferon-stimulated genes. Therefore, there must be a mechanism for inducing the interferon response that is independent of the processing function of RNASET2 for TLR8. However, this situation is not without precedent: the endolysosomal DNase DNase2 also promotes TLR9 activation in the endolysosome while inhibiting activation of cytosolic DNA receptors. In fact, it is rather intriguing that this difference between endosomal and cytosolic receptor activation applies to both DNA and RNA and underscores the importance of the cellular compartment in defining, which nucleic acid species are immunostimulatory. We have rewritten the corresponding paragraph in the introduction.

The molecule has also been suggested to have a role in the processing ribosomal RNA, and in mitochondrial function.

Absolutely. The reported biological functions of RnaseT2 are numerous and confusing. A clear storage phenotype, as described in the zebrafish model, could not be seen in the cells and tissues of the *Rnaset2*^{-/-} mouse. Further investigations are ongoing. However, we do clearly see a type I interferon-dependent phenotype, which results from a deficiency in RNaseT2 activity.

A few thoughts:

The authors state that their model recapitulates "...the devastating, interferon-induced neuroinflammatory encephalopathy found in patients". I think that this is going a bit far – in is not clear that the neurological phenotype in RNASET2 deficiency in humans is interferon driven. It might be, but there is not so much evidence in support of that possibility. I think that they should tone this statement down.

This issue relates to my point above – the authors could use their mouse to provide important evidence as to the basis of the neuropathology of the disease (at least in their mouse) – I don't quite understand why they are publishing now.

Based on the clinical phenotype, which resembles conatal viral infections, as well as the results from this mouse model and indications of interferon elevation in the blood of individual patients, we are convinced that the human phenotype is interferon dependent. However, we agree to the referee that there is a lack of conclusive evidence at this point, so we have toned down this statement as recommended.

Have the authors crossed their mouse with an IFNAR-null mouse? Until they do that, they cannot claim that the neurological phenotype is interferon induced.

Indeed. We were able to crossbreed *Rnaset2*^{-/-} *Ifnar1*^{-/-} mice and have added these data to the manuscript. As mentioned above, the concurrent deficiency in IFNAR alleviates the neuroinflammation observed in the *Rnaset2*^{-/-} mouse.

Have the authors crossed their mouse with a MAVS or STING-null mouse?

We are still elucidating the receptors responsible for the ISG signature observed in the *Rnaset2*^{-/-} mouse by means of CRISPR-experiments *in vitro* as well as by crossbreeding *in vivo*. So far, no meaningful results are available in this regard.

I think that they should be more agnostic about the mechanism until the relevant crosses have been done. For example, perhaps RNASET2 has a role in the ER-Golgi axis on STING signalling. As such, it seems to me that their Figure 8 is pretty much pure speculation.

Thank you for pointing this out. We have deleted the former figure 8 from the revised version of the manuscript. We also have significantly reduced speculative phrasing in the discussion.

Is there any evidence of rRNA accumulation (their reference 16)? What about mitochondria (their reference 19)?

As detailed above, we see no evidence of rRNA accumulation in the *Rnaset2*^{-/-} mouse. Future studies will be necessary –perhaps on further endolysosomal RNases—to understand the species-specific differences observed in this regard.

The mice appear to demonstrate a lethal systemic inflammation: is there any evidence of antibody production and immunological kidney disease (a murine ‘lupus’ like phenotype)?

To answer the question of systemic autoimmunity and antibody production, we have performed indirect immunofluorescence to detect anti-nuclear antibody and ELISA for double-stranded DNA antibodies from serum of *Rnaset2*^{-/-} mice. In addition, a histological assessment of the kidneys and a protein analysis of the urine were carried out.

Rnaset2^{-/-} mice show positive immunofluorescence for anti-nuclear antibody in a high percentage but lack double-stranded DNA antibodies. However, the anti-nuclear antibody in *Rnaset2*^{-/-} mice did not lead to immune complex-mediated nephritis during the observation period of 6 months (see Supplementary Fig. 4).

Reviewer #2: This study describes a new experimental model of RNaseT2 deficiency in mice, which leads to a generalized type1 interferonopathy, which also is associated with CNS neurodegeneration. The topic is interesting, since the induction of a type 1 interferonopathy, which also affects the central nervous system, in a mouse model provides an interesting tool to study not only Aicardi-Goutieres Syndrome (AGS), but may also elucidate basic mechanisms of virus induced neurodegeneration. The study describes in detail the model, the clinical and pathological phenotype and provides clear evidence that RNaseT2 deficiency induces an interferonopathy. Further studies on the pathogenetic mechanisms of neurodegeneration are not provided.

Critique:

A major problem of the study is that it is not fully clear, what is the aim of the study.

It could be the description of the new model and a detailed discussion of its usefulness, its advantages over other models and its limitations. In this respect the study is rather incomplete. Modeling of human RNaseT2 deficiency is rather imperfect. The human disease is predominately a leukodystrophy. Whether there are white matter lesions in the respective model and to what extent they reflect the situation in humans is not addressed in this study.

How the grey matter lesions, described in this study, really compare to those in patients is also not described and discussed in depth. In this respect a table comparing the clinical and pathological phenotype of this model with the human disease and the AGS models would be important.

Originally, the publication was intended as a detailed description of the neuroinflammation observed in a novel mouse model of type I interferonopathy.

It is correct that the *Rnaset2*^{-/-} mouse does not show any leukodystrophy, histologically or morphologically (MRI). At this point, as with the calcifications and cysts, the modeling of the human disease is rather incomplete. However, the calcifications and the formation of cysts are also a variable part of the phenotype in humans. Regarding the grey matter involvement, we can only speculate about their presence in humans due to the lack of autopsy results. RNASET2-related disease is also a gray matter disease as illustrated by the pronounced atrophy and cognitive impairments in humans and mice. In humans, this phenotype often exceeds the spasticity seen in patients. In addition, half of the patients suffer from epileptic seizures.

RNASET2-related phenotypes in humans and mice

Symptom / clinical feature	Humans	Mice
MRI abnormalities:		
- Changes in white matter (leukodystrophy)	++	-
- Brain atrophy	++	++
- Intracranial cystic lesion	(+)	-
- Cerebral calcification	(+)	-
Intellectual disability	++	+
Spastic movement disorder	+	-

Seizures	(+)	-
----------	-----	---

For a better overview of the phenotypes of AGS mouse models, we have added a table of comparison of all known type I interferon mouse models including the *Rnaset2*^{-/-} mouse described here to the supplement.

Furthermore, it would be important to differentiate, how much of the described phenotype is a direct consequence of RNaseT2 deficiency, and what is really triggered by the interferonopathy.

We absolutely agree with the statement. Therefore, we added our very recent results of the *Rnaset2*^{-/-}/*Ifnar1*^{-/-} double knockout mouse to this manuscript. As in figure 8 and the results section "Concurrent deficiency of *Ifnar1* prevents *Rnaset2*^{-/-}-mediated neuroinflammation", we characterize the phenotype of *Rnaset2* deficiency in the absence of type I interferon signaling. Strikingly, the neuroinflammatory phenotype observed for the *Rnaset2*^{-/-}-mouse disappeared, whereas organ size and disturbed hematopoiesis only improves.

Another aim could be to use such a new model to unravel the molecular mechanisms of neurodegeneration. This has been done in part before in another model of Type 1 interferon driven neurodegeneration, which has previously been published in Nature Communications (Rubino et al).

For such a study one would have to identify the cells in the CNS, which drive the lesions, to separate type 1 interferon effects from RNaseT2 deficiency alone, to define the cellular source of type 1 interferons in the CNS and what cells are affected by type 1 IFN signaling.

We agree with the reviewer and have now attempted to create such a model to investigate the underlying mechanism of the observed type I interferon mediated neurodegeneration. One advantage of our model is that it is a whole-body knockout, allowing us to analyze the processes of interferon-mediated neurodegeneration without any artificial changes, such as the diphtheria toxin induced microglia depletion used in Rubino, et al.

To identify which cells are involved in interferon signaling and how these cells change in the course of neuroinflammation, we have performed a single nuclei RNA sequencing analysis. (Please see new figure 7, figure 8 and supplemental figure 7, respectively).

In brief, we could add evidence for type I interferon signaling in all CNS cell types. In addition, the dysregulation of homeostatic functions in astrocytes, oligodendrocytes and OPCs along with markedly alterations of neuronal subclusters provides first insights into potential mechanism of cognitive impairment in context of interferon-driven neuroinflammation.

The authors put much emphasis on the infiltration of the CNS by MHC Class I restricted T-cells, but whether these cells are just secondarily recruited or play a role in the neurodegenerative disease remains unresolved.

Also the differential involvement of microglia versus recruited macrophages and their role in neurodegeneration has not been elucidated.

This is an important point. We absolutely agree that the role of MHC Class I restricted T-cells in the pathomechanism of the observed neuroinflammation as well as the role of microglia cells should be carefully characterized. Thus, we plan to crossbreed the *Rnaset2*^{-/-} mice to Cd8a KO mice. In addition, further experiments with microglia depletion are on the way. However, we think that the results of such attempts should rather be published separately.

In this respect the authors have included Figure 8, depicting a potential pathomechanistic model, but the study does not contain the respective supportive data.

Thank you for pointing this out. We have deleted this hypothetical figure in the revised manuscript.

REVIEWER COMMENTS

Reviewer #1 (Remarks to the Author):

Resubmission

As I said before, I like the basis of this manuscript, and the authors presents a considerable amount of extra work since the original submission. Whilst they do not address the issue that I raised of the signalling pathway involved, the authors do present an Rnaset2/Ifnar cross that suggests a decrease in the neuroinflammation observed in the Rnaset2 model. I have a couple of questions here – I was not completely sure of how many DKO mice had been generated and studied. Apologies if it is my oversight, but I would ask for these data to be completely explicit. Secondly, the DKO mice were culled at 4 months. Why? Why not derive a survival curve so that we can see if the DKO cross savages the whole-animal phenotype?

Reviewer #2 (Remarks to the Author):

Overall, the authors have revised their manuscript and addressed all points raised by the reviewers. Thus, the points of critique are now clarified with this revision. The only caveat is that the very important experiment of Rnaset2 -/- / Ifnar1 -/- double knock out animals is based apparently on a very small animal number (according to Fig 8 only two double knock out animals)and it is not really clear, how a statistically significant result can be obtained on this basis. In addition, it is not clear, whether in the double knock outs also neurodegeneration is absent.

Reviewer #3 (Remarks to the Author):

Using a novel mouse model of RNaseT2 deficiency, the authors demonstrate that RNaseT2 deficiency upregulates type I interferon signaling and results in a neurologic phenotype manifesting both pathologically with evidence of neuroinflammation and immune activation and behaviorally with deficits in memory tasks. Their newly incorporated studies with Rnaset2-/- Ifnar1-/- mice demonstrate that the neuropathologic phenotype is dependent on type I interferon signaling. They further demonstrate using single nuclei RNA sequencing analyses that astrocytes, neurons, oligodendrocytes, and most of all microglia and macrophages in RNaseT2 deficient mice show enhanced type I interferon signaling, among other gene expression alterations. This analysis further identified significant perturbation of homeostatic functions in glial cells (e.g. axon development and ion transport) as well as disturbances in neuronal subpopulations. These additional experiments elevate this study from a purely descriptive study to one beginning to dissect the mechanistic underpinnings of this novel mouse model.

This study is novel as it is the first reported mouse model of RnaseT2 deficiency. This model not only provides a tool for studying neurologic type 1 interferonopathies, but is also useful for dissecting some of the basic mechanisms of viral and autoimmune-induced neuroinflammation, glial pathology, and neurodegeneration. Overall, the study is interesting and important to the fields of neuroimmunology, neuroinfectious disease, neuroinflammation, and CNS innate immune biology. Below are a few comments mainly in reference to the newly provided data and edited text.

Comments

1. The important mechanistic results of the new Ifnar1-/- experiments should be referenced in the abstract. For example, the authors could edit line 50 to “upregulation of interferon-stimulated genes and concurrent Ifnar1-dependent neuroinflammation”
2. The added sentence on line 90-93 in the introduction is confusing as written. Consider re-writing as “Thus while RNaseT2 activity has been shown to induce type I interferon signaling through degradation of longer exogenous RNA molecules into ligands for the pattern recognition receptor TLR8, accumulation of longer endogenous immunostimulatory RNA molecules as a result of

RNaseT2-deficiency may also activate interferon and other innate immune pathways.” While this point is discussed further in the discussion, this is important for readers to understand prior to reading the results of this study.

3. In figures 4 and 5 the N=4 is too low. Was there no replication of this experiment? In figure 4d there are essentially no CD8s in the brain, so in 4e how do the authors find 67% of CD8s are Tem in control brains? Showing flow plots here might be more convincing.

4. In figure 5, the hippocampal atrophy is hard to reconcile with active inflammation. Typically atrophy follows inflammation. Again, N=4 makes this figure weak. I would be more convinced if the authors would show measurements of hippocampi on both sides of the brain and at different time points.

5. The CD8 infiltrates are interesting but it remains unclear whether this is just a secondary phenomenon or if the authors think it's a part of the pathology. Do they have data on chemokine expression or MHC class I>II expression in the CNS to implicate antigen presentation to CD8s?

6. In Figure 8 and its legend, please use *Rnaset2*^{-/-} *Ifnar1*^{-/-} and *Rnaset2*^{-/-} instead of AB KO and ABi knockout. This new nomenclature is not used in other figures or in the result section.

7. In Figure 8B, please provide the standard deviation/error, the number of mice in each group, and statistical analysis for the presented data. Please use scatter plots with mean and standard error similar to Figure 8D and other figures.

8. In Figure 8D, the lack of significance in panel 2 and 3 is seemingly due to the low n (n = 2) in the *Rnaset2*^{-/-} *Ifnar1*^{-/-} mice. Could more double KO mice be included to increase the power of these analyses?

9. Does the knockout of *Ifnar1* in *Rnaset2*^{-/-} mice also reverse/prevent i) the described deficits on memory tasks; ii) the decreased rate of survival; and iii) the hippocampal atrophy? Since the novelty here is the CNS involvement of these mice, this would seem to be an important aspect to show.

10. In figure 7, the KO animals appear to have marked changes in OL and OPC pools, as well. Given recent reports in Nature journals of an immune OL lineage profile in other model systems, it would be interesting and relevant to show gene expression profiles associated with these cells.

11. On line 248 the authors write that they performed single nuclei RNA sequencing analysis on the “caudate putamen (Fig. 7) and hippocampus (Supplementary Fig.7”, however, supplementary figure 7 shows gene expression/biological processes analysis from the caudate/putamen in astrocytes, oligodendrocyte, OPCs, and select neuron clusters. The hippocampus data does not appear to be presented in the supplementary figures. Given the evidence of hippocampal atrophy and memory deficits in their model the single nuclei RNA sequencing and analysis from the hippocampus should be included.

12. The authors report in their response to reviewers “as detailed above, we see no evidence of rRNA accumulation in the *RNase2*^{-/-} mouse”, however, there is not section above that details this data or data on mitochondrial RNA accumulation. Given their proposed mechanism that *RNase2*-deficiency results in type I interferon signaling via accumulation of longer immunostimulatory RNA molecules (possibly rRNA or mtRNA based on prior work in cited references), the accumulation of such RNA molecules should be quantified in these mice and this data should be included in the manuscript.

We would like to thank the reviewers for careful assessment of our work. The comments, suggestions and criticisms stimulated additional data analyses, further experiments and extensive photo documentation, which resulted, in our opinion, in a substantially improved manuscript.

The following additional data have been added to the manuscript:

1. Increased number of animals for calculating survival curve (revised Fig. 1a)
2. Increased number of animals for neuropathology (FACS analysis, revised Fig. 4d)
3. Increased number of animals for *ex vivo* MRI analysis of *Rnaset2*^{-/-} mice including hippocampus (additional ROI) (revised Fig. 5c,d)
4. Increased number of animals for analysis of *Rnaset2*^{-/-}/*Ifnar1*^{-/-} mice (n=5-6)
 - a. Neuropathology (immunohistochemical staining) (revised Fig. 8a,b)
 - b. *Ex vivo* MRI analysis (revised Fig. 8c,d)
 - c. Organ size and blood values (revised Suppl. Fig. 11 a-c)
 - d. Significantly prolonged survival of *Rnaset2*^{-/-}/*Ifnar1*^{-/-} mice (revised Suppl. Fig 11d)
5. Flow chart for CD8⁺ T cell phenotypes (new Suppl. Fig. 6)
6. SnRNAseq data from hippocampus of *Rnaset2*^{-/-} mice (new Suppl. Fig. 8)
7. MHC class I expression data from snRNAseq (new Suppl. Fig. 10a-f)
8. Validation of snRNAseq data for MHC I expression on microglia in FACS analysis (new Suppl. Fig. 10g)

All changes in our revised manuscript are highlighted in yellow. We have addressed the comments to the best of our knowledge and are presenting our response in a point-to-point format.

Reviewer #1 (Remarks to the Author):

As I said before, I like the basis of this manuscript, and the authors presents a considerable amount of extra work since the original submission. Whilst they do not address the issue that I raised of the signalling pathway involved, the authors do present an *Rnaset2*/*Ifnar* cross that suggests a decrease in the neuroinflammation observed in the *Rnaset2* model. I have a couple of questions here – I was not completely sure of how many DKO mice had been generated and studied. Apologies if it is my oversight, but I would ask for these data to be completely explicit.

We appreciate the reviewer's commentary on the relevance of this animal model. We increased the sample size of the *Rnaset2*^{-/-}/*Ifnar1*^{-/-} mice from previously one or two animals to now five or six animals. We can demonstrate that IFNAR1-deficiency abolishes neuroinflammation in all analyzed animals (n=6, revised **Figure 8a, b**) and abrogates hippocampal atrophy in MRI (n=5, revised **Figure 8c, d, e**). Most peripheral organ pathologies were rescued or at least significantly diminished in *Rnaset2*^{-/-}/*Ifnar*^{-/-} mice compared to IFNAR1-competent *Rnaset2*^{-/-} mice (n =5-6, revised Supplemental Figure 11). Thus, IFNAR1-signaling is of utmost importance for the phenotype of *Rnaset2*^{-/-} mice.

Secondly, the DKO mice were culled at 4 months. Why? Why not derive a survival curve so that we can see if the DKO cross savages the whole-animal phenotype?

We fully agree that a “real-world” survival curve of both *Rnaset2*^{-/-} and *Rnaset2*^{-/-}*Ifnar1*^{-/-} mice would be highly desirable. In the *Rnaset2*^{-/-} strain some animals died spontaneously, but many mice had to be sacrificed at the request of the veterinarians in order to attain the pre-specified endpoint criteria for animal welfare reasons (death of animals is not an accepted endpoint for animal experiments in Germany).

Of note, these decisions were made on a clinical basis by animal caretakers and veterinarians unaware of the genotype, which is now explicitly stated in the manuscript (see line 145-150).

None of the *Rnaset2*^{-/-}*Ifnar1*^{-/-} mice up to the age of 6 months have died spontaneously or had to be sacrificed at the request of the caretakers or veterinarians. With the above described limitations, we can demonstrate a significantly increased life expectancy of *Rnaset2*^{-/-}*Ifnar1*^{-/-} mice compared to *Rnaset2*^{-/-} mice (see Supplemental Figure 11d).

Reviewer #2 (Remarks to the Author):

Overall, the authors have revised their manuscript and addressed all points raised by the reviewers. Thus, the points of critique are now clarified with this revision.

The only caveat is that the very important experiment of *Rnaset2*^{-/-} / *Ifnar1*^{-/-} double knock out animals is based apparently on a very small animal number (according to Fig 8 only two double knock out animals) and it is not really clear, how a statistically significant result can be obtained on this basis. In addition, it is not clear, whether in the double knock outs also neurodegeneration is absent.

We now included four additional *Rnaset2*^{-/-}*Ifnar1*^{-/-} mice in the analysis and provide compelling evidence that IFNAR1-deficiency abolishes neuroinflammation (revised Figure 8a, b, c) (see also our response to reviewer #1). Furthermore, hippocampal atrophy also resolved in *Rnaset2*^{-/-}*Ifnar1*^{-/-} mice as shown by *ex vivo* MRI (revised Figure 8d, e).

Reviewer #3 (Remarks to the Author):

Using a novel mouse model of RNaseT2 deficiency, the authors demonstrate that RNaseT2 deficiency upregulates type I interferon signaling and results in a neurologic phenotype manifesting both pathologically with evidence of neuroinflammation and immune activation and behaviorally with deficits in memory tasks. Their newly incorporated studies with *Rnaset2*^{-/-} *Ifnar1*^{-/-} mice demonstrate that the neuropathologic phenotype is dependent on type I interferon signaling. They further demonstrate using single nuclei RNA sequencing analyses that astrocytes, neurons, oligodendrocytes, and most of all microglia and macrophages in RNaseT2 deficient mice show enhanced type I interferon signaling, among other gene expression alterations. This analysis further identified significant perturbation of homeostatic functions in glial cells (e.g. axon development and ion transport) as well as disturbances in neuronal subpopulations. These additional experiments elevate this study from a purely descriptive study to one beginning to dissect the mechanistic underpinnings of this novel mouse model.

This study is novel as it is the first reported mouse model of RnaseT2 deficiency. This model not only provides a tool for studying neurologic type 1 interferonopathies, but is also useful for dissecting some of the basic mechanisms of viral and autoimmune-induced neuroinflammation, glial pathology, and neurodegeneration. Overall, the study is interesting and important to the fields of neuroimmunology, neuroinfectious disease, neuroinflammation, and CNS innate immune biology. Below are a few comments mainly in reference to the newly provided data and edited text.

Comments

1. The important mechanistic results of the new *Ifnar1*^{-/-} experiments should be referenced in the abstract. For example, the authors could edit line 50 to “upregulation of interferon-stimulated genes and concurrent *Ifnar1*-dependent neuroinflammation”

We very much appreciate this suggestion and have rewritten the abstract accordingly (see line 52).

2. The added sentence on line 90-93 in the introduction is confusing as written. Consider re-writing as “Thus while RNaseT2 activity has been shown to induce type I interferon signaling through degradation of longer exogenous RNA molecules into ligands for the pattern recognition receptor TLR8, accumulation of longer endogenous immunostimulatory RNA molecules as a result of RNaseT2-deficiency may also activate interferon and other innate immune pathways.” While this point is discussed further in the discussion, this is important for readers to understand prior to reading the results of this study.

We have rewritten this paragraph accordingly to emphasize the various aspects of the RNaseT2 functions more clearly (see line 91-98).

3. In figures 4 and 5 the N=4 is too low. Was there no replication of this experiment?

We replicated the experiments and increased the number of analyzed animals in both figures (n=8, see revised Figure 4d and, revised Figure 5c). We also report now that the number of CD4⁺ T cells and CD19⁺ B cells is significantly higher in the CNS of *Rnaset2*^{-/-} mice compared to controls by FACS analysis (see line 219-221).

In figure 4d there are essentially no CD8s in the brain, so in 4e how do the authors find 67% of CD8s are Tem in control brains? Showing flow plots here might be more convincing.

We made now individual graphs for each cell type and split the y-axis in all graphs (revised Figure 4d). In addition, we provide an exemplary flow cytometry dot plot as Supplemental Figure 6.

4. In figure 5, the hippocampal atrophy is hard to reconcile with active inflammation. Typically atrophy follows inflammation. Again, N=4 makes this figure weak.

We agree with the referee that inflammation usually precedes brain atrophy and can mask the latter. We could now add four additional *Rnaset2*^{-/-} and control mice to the *ex vivo* MRI analysis (see revised Figure 5c, d). In addition, we now provide a coronar MR image of the hippocampus and quantitated the hippocampal T2 relaxation times in both genotypes (see revised Figure 5d). We report that both, the T2-relaxation time of all analyzed brain areas (see revised Figure 5d) and the hippocampal atrophy (see

revised Figure 5c) are significantly increased in *Rnaset2*^{-/-} compared to control mice, indicating that at the time points studied, evidence of both inflammation and atrophy are encountered. Importantly, T2 hyperintensity as well as brain atrophy were completely rescued in *Rnaset2*^{-/-}*Ifnar1*^{-/-} mice, clearly indicating that excess IFNAR1-signalling both instigates neuroinflammation and brain atrophy (see **revised Figure 8c, d, e**).

I would be more convinced if the authors would show measurements of hippocampi on both sides of the brain and at different time points.

We quantified the size of the hippocampus in both hemispheres. In **Figure 5c** we compared the sum of both hippocampal areas in relation to the total brain area per mouse between *Rnaset2*^{-/-} and control mice. Similar results were obtained when we compared the hippocampal areas of the right or left hemisphere separately between the genotypes. To make this clear to the reader, we added corresponding arrows on both sides in **Figure 5b** and changed the figure legend to: “Percent area of hippocampus (*left and right*) with respect to whole brain area” (see **line 1046-1047**).

In the frame of this study, we did not perform repetitive MRI measurements. However, in the data set available so far, we did not observe an impressive increase in hippocampal atrophy in *Rnaset2*^{-/-} mice comparing 6- versus 4-month-old animals.

5. The CD8 infiltrates are interesting but it remains unclear whether this is just a secondary phenomenon or if the authors think it's a part of the pathology. Do they have data on chemokine expression or MHC class I>II expression in the CNS to implicate antigen presentation to CD8s?

This point is well taken. Although likely, we have at present no definite proof that inflammation or CD8⁺ T cells in particular underlie the observed brain atrophy in *Rnaset2*^{-/-} mice. Interestingly, however, IFNAR1-deficiency strikingly abolishes not only the neuroinflammatory phenotype but also hippocampal atrophy, demonstrating that there is no atrophy independent of IFNAR1 suggesting that neuroinflammation and neurodegeneration may indeed be tightly interconnected.

In this regard, we also addressed whether MHC class I is preferentially upregulated and analyzed MHC class I and II expression side-by-side. Interestingly microglia (and all other analyzed CNS cells) of *Rnaset2*^{-/-} mice did not show an upregulated MHC class II expression if compared to control microglia. This is in stark contrast to the pronounced upregulation of MHC class I that we found in all CNS cells studied (**Supplemental Figure 10a-f**, added result section **line 310-314**). We also confirmed the increased MHC-I expression on microglia in *Rnaset2*^{-/-} mice by flow cytometry (**Supplemental Figure 10g**, added result section **line 314-316**).

With regard to chemokine expression, we observed an upregulation of CCL5/RANTES on multiple cell clusters (including microglia and neurons) and an upregulation of CCL12 exclusively on the microglia cluster of *Rnaset2*^{-/-} mice compared to controls. CCL5 is a well-known chemokine for CD8⁺ T cells.¹

Thus, one might speculate that the CNS of *Rnaset2*^{-/-} mice is well equipped to recruit and expand CD8⁺ T cells. We changed a paragraph of the discussion accordingly (see **line 373-381**):

¹Thierry Walzer, Antoine Marçais, Frédéric Saltel, Chantal Bella, Pierre Jurdic and Jacqueline Marvel. Cutting Edge: Immediate RANTES Secretion by Resting Memory CD8 T Cells Following Antigenic Stimulation. *J Immunol* **170** (4) 1615-1619.

6. In Figure 8 and its legend, please use *Rnaset2*^{-/-} *Ifnar1*^{-/-} and *Rnaset2*^{-/-} instead of AB KO and ABI knockout. This new nomenclature is not used in other figures or in the result section.

We apologize and use now the nomenclature *Rnaset2*^{-/-} and *Rnaset2*^{-/-}*Ifnar1*^{-/-} consistently throughout the manuscript.

7. In Figure 8B, please provide the standard deviation/error, the number of mice in each group, and statistical analysis for the presented data. Please use scatter plots with mean and standard error similar to Figure 8D and other figures.

We have changed **Figure 8b** as suggested.

8. In Figure 8D, the lack of significance in panel 2 and 3 is seemingly due to the low n (n = 2) in the *Rnaset2*^{-/-} *Ifnar1*^{-/-} mice. Could more double KO mice be included to increase the power of these analyses?

We appreciate the commentary of the referee and added four additional *Rnaset2*^{-/-}*Ifnar1*^{-/-} mice to increase the power (see **added Supplemental Figure 11c**). We were able to discover significant differences between the genotypes. IFNAR1-deficiency normalized Hb, MCV and MCH of *Rnaset2*^{-/-}*Ifnar1*^{-/-} mice to control levels. The elevated WBC numbers were significantly decreased and the prominently reduced platelet numbers were increased in *Rnaset2*^{-/-}*Ifnar1*^{-/-} compared to *Rnaset2*^{-/-} mice, but did not normalize completely.

9. Does the knockout of *Ifnar1* in *Rnaset2*^{-/-} mice also reverse/prevent i) the described deficits on memory tasks; ii) the decreased rate of survival; and iii) the hippocampal atrophy? Since the novelty here is the CNS involvement of these mice, this would seem to be an important aspect to show.

We fully agree on the importance of these additional questions raised by the referee. We were able to demonstrate that neuroinflammation and hippocampal atrophy were both abolished in *Rnaset2*^{-/-}*Ifnar1*^{-/-} mice (see **revised Figure 8**). Unfortunately, however, we did not obtain enough *Rnaset2*^{-/-}*Ifnar1*^{-/-} mice of equal sex to perform the behavioral experiments as with *Rnaset2*^{-/-} mice. We could, however, demonstrate that life expectancy is significantly prolonged in *Rnaset2*^{-/-}*Ifnar1*^{-/-} compared to *Rnaset2*^{-/-} mice (see **added Supplemental Figure 11d**).

10. In figure 7, the KO animals appear to have marked changes in OL and OPC pools, as well. Given recent reports in Nature journals of an immune OL lineage profile in other model systems, it would be interesting and relevant to show gene expression profiles associated with these cells.

Upregulated interferon type I stimulated genes, especially *Ddx60* and raised transcripts of MHC class I were the most prominent transcriptional changes in the oligodendrocyte and OPC clusters of *Rnaset2*^{-/-} mice compared to controls (see **Supplemental Figure 8d**). MHC class I transcripts of the caudate putamen and the hippocampus are now displayed in **Supplemental Figure S10** and fit well with the OL lineage profile of Falcão et al.² during EAE. MHC-II expression in OL lineage cells of *Rnaset2*^{-/-} mice compared to controls was unchanged and thus different from EAE mice in which OL lineage cells upregulated MHC-II.

²Ana Mendanha Falcão; David Bruggen; Sueli Marques; Mandy Meijer; Sarah Jäkel; Eneritz Agirre; Samudyata; Elisa M. Floriddia; Darya P. Vanichkina; Charles ffrench-Constant; Anna Williams; André Ortlieb Guerreiro-Cacais; Gonçalo Castelo-Branco. Disease-specific oligodendrocyte lineage cells arise in multiple sclerosis. *Nat Med* **24**, 1837–1844 (2018).

11. On line 248 the authors write that they performed single nuclei RNA sequencing analysis on the “caudate putamen (Fig. 7) and hippocampus (Supplementary Fig.7”, however, supplementary figure 7 shows gene expression/biological processes analysis from the caudate/putamen in astrocytes, oligodendrocyte, OPCs, and select neuron clusters. The hippocampus data does not appear to be presented in the supplementary figures. Given the evidence of hippocampal atrophy and memory deficits in their model the single nuclei RNA sequencing and analysis from the hippocampus should be included.

We now included the analysis of the single nuclei RNA sequencing from the hippocampus as **Supplemental Figure 8**. The corresponding description can be found in the results section (see **line 298-309**).

Data availability

Raw single nuclei RNA sequencing data are accessible from the NCBI GEO database using the accession number GSE180138. <https://www.ncbi.nlm.nih.gov/geo/query/acc.cgi?acc=GSE180138>

Until publication, please use the following reviewer token: **azijsyqklfgljqd**

12. The authors report in their response to reviewers “as detailed above, we see no evidence of rRNA accumulation in the RNase2^{-/-} mouse”, however, there is not section above that details this data or data on mitochondrial RNA accumulation. Given their proposed mechanism that RNase2-deficiency results in type I interferon signaling via accumulation of longer immunostimulatory RNA molecules (possibly rRNA or mtRNA based on prior work in cited references), the accumulation of such RNA molecules should be quantified in these mice and this data should be included in the manuscript.

We absolutely agree with the referee that it would be relevant to demonstrate that RNA accumulates in *Rnaset2*^{-/-} mice.

So far, we did not succeed in obtaining a convincing IHC staining for RNA in the CNS tissue using the 9D5 antibody. However, preliminary data show a trend for raised mitochondrial and ribosomal RNA by qPCR analysis of mouse embryo fibroblasts derived from *Rnaset2*^{-/-} mice compared to controls (see figure below, n=7 biological repeats, Calr as housekeeping gene).

We thank the reviewers again for their thorough assessment of our work and hope that we could answer the questions that arose during the review process.

REVIEWER COMMENTS

Reviewer #1 (Remarks to the Author):

Well done to the authors - a very interesting study.

Reviewer #2 (Remarks to the Author):

The authors have addressed and clarified all points raised by the reviewers in the revision of the manuscript.

Reviewer #3 (Remarks to the Author):

Overall, the authors have been quite responsive to my comments, particularly their inclusion of significantly more *Rnaset2*^{-/-} *Ifnar*^{-/-} mice and adjustment of data presentation that was suggested. I think it is acceptable that they did not conduct the behavioral studies on the *Rnaset2*^{-/-} *Ifnar*^{-/-} mice.

My only remaining question/critique is really still point 12 regarding the hypothesized mechanism by which RNaseT2 deficiency leads to activation of interferon signaling. They still hypothesize in the introduction that RNaseT2 deficiency may lead to “the accumulation of long endogenous immunostimulatory RNA molecules may also activate interferon and other innate immune pathways,” but they have no data to support this in their manuscript. I think this level of mechanism given their inclusion of the *Ifnar*^{-/-} data could be the subject of a future mechanistic study though and while nice I don't think would be required for this study.

Reviewer #3 (Remarks to the Author):

Overall, the authors have been quite responsive to my comments, particularly their inclusion of significantly more Rnaset2^{-/-} Ifnar^{-/-} mice and adjustment of data presentation that was suggested. I think it is acceptable that they did not conduct the behavioral studies on the Rnaset2^{-/-} Ifnar^{-/-} mice.

My only remaining question/critique is really still point 12 regarding the hypothesized mechanism by which RNaseT2 deficiency leads to activation of interferon signaling. They still hypothesize in the introduction that RNaseT2 deficiency may lead to “the accumulation of long endogenous immunostimulatory RNA molecules may also activate interferon and other innate immune pathways,” but they have no data to support this in their manuscript. I think this level of mechanism given their inclusion of the Ifnar^{-/-} data could be the subject of a future mechanistic study though and while nice I don’t think would be required for this study

We agree with the comment of reviewer #3 and have removed the criticized hypothesis from our introduction (line 101f).